# Whole-rock and zircon evidence for evolution of the Late Jurassic high Sr/Y Zhoujiapuzi granite, Liaodong Peninsula, North China Craton

**Renyu Zeng** [a, b,c] *, **Mark B. Allen** [c], **Xiancheng Mao** [d], **Jianqing Lai** [d], **Jie Yan** [a, b], **Jianjun Wan**[a, b]

[a] State Key Laboratory of Nuclear Resources and Environment, East China University of Technology, Nanchang, 330013, Jiangxi, China

[b] School of Earth Sciences, East China University of Technology, Nanchang, 330013, China

[c] Department of Earth Sciences, Durham University, Durham DH1 3LE, UK

[d] School of Geosciences and Info-Physics, Central South University, Changsha 410083, China

**Correspondence:** Renyu Zeng (zengrenyu@126.com)

**Abstract**: Middle-Late Jurassic high Sr/Y granitic intrusions are extensively exposed in the Liaodong Peninsula, in the eastern part of the North China Craton (NCC). However, the genesis of the high Sr/Y signature in these intrusions has not been studied in detail. In this study, we report results of zircon U-Pb dating, Hf isotopic analysis and zircon and whole-rock geochemical data for the Late Jurassic Zhoujiapuzi granite in the middle part of the Liaodong Peninsula. The Zhoujiapuzi granite is high-K (calc-alkaline) and peraluminous in nature, with high $SiO_2$ (68.1–73.0 wt %) and $Al_2O_3$

(14.5–16.8 wt %), low in $TFe_2O_3$ (1.10–2.49 wt %) and $MgO$ (0.10–0.44 wt %), and
with high Sr/Y (19.9–102.0) and $La_N/Yb_N$ (14.59–80.40), characteristic of high Sr/Y I-
type granite. The geochemical signatures, in combination with the presence of a large
number of Paleoproterozoic inherited zircons, indicate that the Zhoujiapuzi granite was
most likely derived from partial melting of the basement in the region, and specifically
the Liaoji granites. The high Sr/Y signature is inherited from these source rocks. LA-
ICP-MS zircon U-Pb dating of the autocryst zircons from two samples (from different
localities) yielded consistent weighted average ages of 160.7±1.1 Ma (MSWD=1.3) and
159.6±1.1 Ma (MSWD=1.2), with εHf(t) values in the range of -26.6– -22.8.
Morphological and chemical studies on autocrystic zircon grains show that there are
two stages of zircon growth, interpreted as magmatic evolution in two distinct stages.
The light-CL core reflects a crystallization environment of low oxygen fugacity and
high $T_{Zr-Ti}$; the dark-CL rim formed with high oxygen fugacity and lower $T_{Zr-Ti}$. Based
on the geochemical features and regional geological data, we propose that the Liaodong
Peninsula in the Late Jurassic was part of a mature continental arc, with extensive
melting of thick crust above the Paleo-Pacific subduction zone.
**Keywords:** Liaodong Peninsula; Late Jurassic; Zircon U-Pb-Hf isotopes; Two stages
of crystal growth; High Sr/Y granite
**1. Introduction**
The Liaodong Peninsula is located in the northeast of the North China Craton
(NCC). The northeast NCC was influenced by three main tectonic regimes in the
Mesozoic, related to the subduction of the Paleo-Asian, Paleo-Pacific and Mongol-
Okhotsk oceans (Tang et al., 2018). The superposition of these different regimes
resulted in changing tectonic and magmatic patterns over time. Middle-Late Jurassic
granitic rocks are extensively exposed in the northern parts of the Liaodong Peninsula,
such as the Yutun mylonitic granite, Xiaoheishan granodiorite, Heigou monzogranite
(Wu et al., 2005), Wulong two-mica monzogranite (Yang et al., 2018), and Huangdi
biotite monzogranite (Xue et al., 2020). Most of these rocks are characterized by high
Sr /Y, and plot within the adakite field on Sr/Y-Y and $La_N/Yb_N$-$Yb_N$ diagrams (Wu et
al., 2005a; Yang et al., 2015a, 2018).
The geodynamic settings and petrogenesis of adakite and geochemically similar
high Sr/Y igneous rocks have been widely discussed. The high Sr/Y rocks were
originally proposed to be formed by melting of young (<25 Ma) and hot subducted
oceanic slab in an arc setting (Defant and Drummond, 1990). However, later studies
have shown that the high Sr/Y rocks can form in both arc and non-arc settings by other
processes, such as continental interior settings (Wang et al., 2007), cold subduction
zones (Nakamura and Iwamori, 2013), collision or post-collision processes (Schwartz
et al., 2011). In addition, numerous studies have suggested that the lower continental
crust can also be the source of the high Sr/Y rocks (Gao et al., 2004; Ou et al., 2017).
However, it is debated whether crustal thickening is necessary for their formation (e.g.
Moyen, 2009; Kamei et al., 2009; Zhan et al., 2020). In recent years, some studies have
proposed that the high Sr/Y ratio in granitic rocks can be inherited from a high Sr/Y
crust source, regardless of pressure (Kamei et al., 2009; Ma et al., 2015; Zhan et al.,

66 2020).

The Middle-Late Jurassic granitic rocks in the Liaodong Peninsula are commonly
proposed to be the products of partial melting of thickened mafic crust with garnet in
the residue (Wu et al., 2005a; Yang et al., 2015a, 2018; Tang et al., 2018). However, the

source composition has not been fully considered in the petrogenesis of the high Sr/Y

rocks in the Liaodong Peninsula. Hence, the petrogenesis of the Middle-Late Jurassic

high Sr/Y rocks needs to be re-evaluated, based on more detailed work and a

consideration of possible sources. This petrogenesis is of significance for understanding

the Jurassic tectonics of the Liaodong Peninsula, and the NCC in general.

In this paper, we examined the high Sr/Y Zhoujiapuzi granite from the Xiuyan area,

in the middle of the Liaodong Peninsula. Zircons are analysed for U-Pb-Hf isotopes

and trace element geochemistry, and by Raman spectroscopy. These results are

integrated with whole-rock geochemistry. We focus on the zircons, because of their

potential to reveal the origins of the pluton (Belousova et al., 2002; Wang et al., 2007;

Breiter et al., 2014; Zhao et al., 2014), and so provide a case study for the evolution of

plutonic magma systems in general. Based on observations of the CL images and

chemical analysis, two zircon growth stages can be distinguished. We first determine

the crystallization environments of the two zircon growth stages, and then decipher the

petrogenesis, source characteristics and origin of the high Sr/Y signature of the pluton

as a whole. Integrated with previous studies, our study provides insights into the

tectonic evolution of the Liaodong Peninsula in the Late Jurassic.

**2. Geological setting**

The Zhoujiapuzi granite is located in the middle of the Liaodong Peninsula, at the

northeastern margin of the NCC (Fig. 1). The Paleoproterozoic Liaohe Group and Liaoji

granite are the basement in the study area. The Liaohe Group includes the Lieryu,

Gaojiayu, Dashiqiao and Gaixian formations. Although stratigraphic terms are used,

these rocks are metamorphic, and the group consists of leptynite, leptite, granulite,

amphibolite, marble and phyllite. The protoliths of the Liaohe Group include marine volcanics, clastics, carbonates and claystones. The formation age of the metasedimentary rocks in the Liaohe Group is 2.0–1.9 Ga (Wan et al., 2006; Li et al., 2015). It is in unconformable contact with the overlying strata of the Mesoproterozoic Cuocaogou Formation and Xiaoling Formation.

The study area experienced strong magmatic activity in the Paleoproterozoic, which can be divided into two stages of 2.2–2.1 Ga and ~ 1.85 Ga. The 2.18–2.14 Ga Liaoji granites (also called gneissic granites), which lie within an area measuring 300 km × 70 km, are dominated by A- and I-type granites (Li and Zhao, 2007; Yang et al., 2016; Wang et al., 2020a). Metamorphosed volcanic rocks (leptynite, leptite and granulite) in the Liaohe Group also formed at 2.2–2.1 Ga (Li et al., 2015). The ~1.85 Ga granites mainly consist of I- and S-type porphyry granites and alkaline syenites (Yang et al., 2007; Yang et al., 2015b). In addition, there were small amounts of mafic magmatic activity at ~2.17 Ga, ~2.1 Ga and ~1.8 Ga (Meng et al., 2014; Yuan et al., 2015). There are a variety of viewpoints on the Paleoproterozoic tectono-magmatic evolution in the Liaodong Peninsula, such as an intracontinental rift opening-closing model (Li et al., 2005) and an arc-continent collision model (Faure et al., 2004).

In the Mesozoic, the region of the Liaodong Peninsula was influenced by the circum-Pacific tectonic regime, the Mongol-Okhotsk tectonic regime and the Paleo-Asian Ocean tectonic regime. The joint influence of multiple tectonic regimes resulted in intensive magmatism during the Mesozoic (Fig. 1b). These Mesozoic magmatic rocks can be divided into three stages, namely: Triassic (233–212 Ma), Jurassic (180–

156 Ma) and Early Cretaceous (131–117 Ma) (Wu et al., 2005b).
The Triassic magmatic rocks are less exposed, mainly alkaline rocks, diabase,
diorites and granites (Wu et al., 2005b). Among them, the granites mainly have A-type
affinity, and may have formed in an extensional setting (Tang et al., 2018; Wang et al.,
2019). Magmatism has been related to either the subduction of the Paleo-Pacific slab,
closure of the Paleo-Asian Ocean, or the collision between the NCC and the Yangtze
Craton (Tang et al., 2018; Wang et al., 2019). The majority of the Jurassic magmatic
rocks are monzogranite and granodiorite, which are generally calc-alkaline I-type
granites, and show characteristics of adakite-like rocks. Some of them, exposed near
later extensional structures, have undergone regional ductile deformation. These
Jurassic magmatic rocks are generally considered to relate to the subduction of the
Paleo-Pacific slab (Wu et al., 2005a; Zhai et al., 2004). In the Early Cretaceous, basic-
acidic-alkaline rocks were widely developed. Among them, the granites have mainly
A- and I-type affinities. These rocks are generally considered to have formed in an
intense extensional environment, which is connected with either the rollback or low-
angle subduction of the Paleo-Pacific slab (Wu et al., 2005c; Zheng et al., 2018).
**3. Samples and petrography**
The Zhoujiapuzi granite is located to the east of Xiuyan City, in the middle of the
Liaodong Peninsula (Fig. 1b). It intruded into the Lieryu Formation of the Liaohe Group.
Eight samples of the Zhoujiapuzi granite were collected at locations shown in Fig. 1c.
The Zhoujiapuzi granite is generally grey in colour and with fine-grained texture
(Fig. 2a). The mineral assemblage contains K-feldspar (~50 %), quartz (~25 %),
plagioclase (~20 %) and biotite (~5 %) as well as accessory minerals such as zircon,
ilmenite, magnetite and apatite. K-feldspar grains are euhedral or subhedral, and always
exhibit cross-hatched twinning (Fig. 2b). Quartz grains are usually xenomorphic, and
have indented boundaries and wavy extinction (Fig. 2b-d). Plagioclase always exhibits
polysynthetic twinning and have sericitization in places (Fig. 2c). Biotite mainly fills
in the interstices between the other minerals (Fig. 2c, d).
**4. Analytical methods**
The cathodoluminescence (CL) images of zircon were obtained by the Chengpu
geological Testing Co. Ltd, Langfang, China using the TIMA analysis. The LA-ICP-
MS zircon U-Pb analyses were performedusing an Agilent Technologies 7700x ICP-
MS with a Teledyne Cetac Technologies Analyte Excite laser-ablation system at
Nanjing FocuMS Contract Testing Co. Ltd. The analyses were carried out with a 35 μm
spot size at 8 Hz repetition rate for 40 seconds. The ICP-MS detector has dual modes:
pulse for lower signal, and analog for higher signal. Pulse-analog cross calibration was
performed before the measurement of U-Pb isotopes, delivering a wider linear dynamic
range – up to 10 orders of magnitude. For a signal of $^{238}$U higher than 1.2–1.4 Mio cps,
equivalent zircon contains U concentrations higher than 600 ppm, and are measured in
analog mode. 91500 was used as external standard. GJ-1 (600Ma, Jackson et al., 2004)
and Plešovice (337Ma, Sláma et al., 2008) were treated as quality control for
geochronology. During our analyses, the weighted mean age of GJ-1 and Plešovice
were 606.0 ± 4.8 Ma (n=16, MSWD = 0.50) and 340.9 ± 4.0 Ma (n=7, MSWD = 1.0),
respectively. Trace elements abundance of zircon were externally calibrated against

NIST SRM 610 with Si as the internal standard. The raw ICP-MS data were processed using ICPMSDataCal software (Liu et al., 2010). No common-Pb correction was applied to the data. Data reduction was completed using the Isoplot4.15 (Ludwig, 2003). The instrument description and analytical procedure are described in detail by Zeng et al. (2018).

The in-situ Lu-Hf isotopic analyses of zircon were performed by LA-MC-ICP-MS using a Teledyne Cetac laser-ablation system and a Nu Plasma II MC-ICP-MS at Nanjing FocuMS Contract Testing Co. Ltd. The 193 nm ArF excimer laser was focused on zircon surface with fluence of $6.0J/cm^2$. The ablation protocol employed a spot diameter of 50 um at 8 Hz repetition rate for 40 seconds. Three standard zircons, GJ-1, 91500, and Penglai, were analysed for quality control at every ten unknown samples. In the experiment, standard zircon GJ-1, 91500, and Penglai were analyzed, and the $^{176}Hf/^{177}Hf$ ratios were 0.282002–0.282013, 0.282305–0.282315 and 0.282901–0.282914 respectively, in accordance with their recommended values (GJ-1: 0.282012, Yuan et al., 2008; 91500: $0.282307 \pm 0.000031$, Wu et al., 2006; Penglai: $0.282906 \pm 0.000010$, Li et al., 2010). For the calculation of εHf(t) values, we have adopted the $^{176}Lu$ decay constant of $1.867 \times 10^{-11}$ (Söderlund et al., 2004), the present-day chondritic values of $^{176}Lu/^{177}Hf = 0.0332$ and $^{176}Hf/^{177}Hf = 0.282772$ (Blichert-Toft and Albarède 1997). To calculate one-stage model ages ($T_{DM1}$) relative to a depleted-mantle source, we have adopted the present-day depleted-mantle values of $^{176}Lu/^{177}Hf = 0.0384$ and $^{176}Hf/^{177}Hf = 0.28325$ (Vervoort and Blichert-Toft 1999). To calculate two-stage modal ages (TDM2), 'felsic crust' model ages are calculated using average continental crust $^{176}Lu/^{177}Hf = 0.015$ (Griffin et al., 2004)

Zircon Raman analyses were carried out using an RM2000 laser Raman
spectrometer at the State Key Laboratory of Nuclear Resources and Environment, East
China University of Technology. The selected incident wavelengths were 532 and 785
nm in order to clearly identify the luminescence bands due to low concentration
impurities. The beam power was 20 mW. The Leica 50× objective was employed.
Six fresh rock samples were selected for geochemical analysis. The elemental
analyses were conducted at Analytical Chemistry & Testing Services (ALS) Chemex
(Guangzhou) Ltd. Major oxides were analyzed using wave-dispersive X-ray
fluorescence (XRF) (ME-XRF26). Analytical precision was better than $\pm$ 0.01%.
Trace element abundances were measured by the lithium borate dissolution method and
ICP-MS (ME-MS81). The analytical uncertainties of the rare earth element (REE) and
high field strength element (HFSE) are <5%. Analytical uncertainties are in the range
of 5%–10% for the other elements. Detailed analytical procedures refer to Zhang et al.
(2019) and Nash et al. (2020).
**5. Analytical results**
The data for major and trace elements, Raman microprobe data, zircon trace
elements, zircon U-Pb ages, and zircon Hf isotopes are shown in Tables S1, S2, S3, S4
and S5, respectively.
**5.1. Whole-rock major and trace element compositions**
SiO$_2$ contents range from 68.11 wt.% to 73.02 wt.% (average 71.71 wt.%).
Contents of Na$_2$O and K$_2$O are 3.81 – 4.65 wt.% and 4.32 – 4.71 wt.%, respectively,
with Na$_2$O/K$_2$O ratio of 0.82 – 1.08 and total alkalis (Na$_2$O + K$_2$O) of 8.38 – 8.97. All
samples plot in the field of granite in the TAS classification except one (Fig. 3a). These
samples have Al$_2$O$_3$ contents of 14.49 – 16.83 wt.% (average 15.09 wt.%), CaO
contents of 1.04 – 1.98 wt.% (average 1.38 wt.%) and A/CNK values of 1.05 – 1.10
(average 1.07). In the A/NK – A/CNK diagram (Fig. 3b), all samples plot in the
peraluminous field (Fig. 3b). The granite samples have low TFe$_2$O$_3$ (TFe$_2$O$_3$ = all Fe
calculated as Fe$_2$O$_3$) contents and MgO contents ranging from 1.10–2.49 wt % and
0.10–0.44 wt %, respectively, with Mg# (Mg#=100*molar Mg/(Mg+Fe)) values of 15–

211 26.

The samples of the Zhoujiapuzi granite exhibit variable REEs, with total REEs
ranging from 59 to 302 ppm. The La$_N$/Yb$_N$ values of the Zhoujiapuzi granite range from
14.59 to 80.40 (average 38.27), showing right-declined REE patterns (Fig. 4a). The
samples have Eu/Eu* of 0.62–1.94 and Ce/Ce* of 0.94–1.16. In the primitive mantle-
normalized trace element diagram (Fig. 4b), the samples show negative anomalies of
HFSEs (e.g., Nb, Ta, Ti and P) and positive anomalies of La and LILEs (e.g., K, Rb,
Ba, U, La, Ce). The Zhoujiapuzi granite is characterized by high contents of Sr (309–
551 ppm) and low contents of Y (5.01–15.5 ppm) and Yb (0.43–1.40 ppm), with high
Sr/Y ratios of 19.94–102.04 (average 65.50).
**5.2 Zircon CL images, Raman spectra and REE elements**
CL images of zircons from the Zhoujiapuzi granite are shown in Fig. 5. Zircons
commonly have crystal sizes between 150 and 250 μm, and have length/width ratios of
2:1–4:1, with euhedral, stubby to elongate prisms. According to the CL images, most
zircons show an internal division into 2 distinct domains: light-CL core and dark-CL

rim. The light-CL core is characterized by bright CL intensity and widely-spaced oscillatory zoning patterns. The dark-CL rim is overgrown continuously by the light-CL core and is characterized by extremely low CL emission and narrowly-spaced oscillatory zoning patterns. In addition, some zircons have inherited cores, which have corroded and rounded shapes in contact with the light-CL core, such as 1# and 37# in XY-001 and 6# and 41# in XY-008 (Fig. 5). These inherited zircons have oscillatory zoning in CL images.

Six light-CL core spots and six dark-CL rim spots were analyzed for Raman spectra. The light-CL cores have antisymmetric stretching vibration ($B_{1g}$) of the $SiO_4$ tetrahedra ($v_3$ ($SiO_4$)) Raman band of 1005–1007 cm$^{-1}$ and half-width of the $v_3$ ($SiO_4$) Raman band ($b$) values of 6.0–8.1 cm$^{-1}$, while the dark-CL rims have $v_3$ ($SiO_4$) Raman band of 1004–1007 cm$^{-1}$ and $b$ values of 5.4–9.0 cm$^{-1}$.

Twenty light-CL core spots, eighteen dark-CL rim spots and six inherited zircon spots were analyzed for trace and rare earth elements. The light-CL core spots have lower U content (28–677 ppm) than the dark-CL rim spots (U=641–3842 ppm). In the chondrite-normalized REE element diagram (Fig. 6a, b), both the light-CL core and dark-CL rim are characterized by HREE enrichment relative to LREE with positive Ce anomalies and negative Eu anomalies. The light-CL core spots have ΣREE of 49–1115 ppm (average 390 ppm), ΣLREE of 3–72 ppm (average 14 ppm) and ΣHREE of 46–1100 ppm (average 377 ppm), whereas the dark-CL rim spots have ΣREE of 327–1632 ppm (average 895 ppm), ΣLREE of 2–14 ppm (average 6 ppm) and ΣHREE of 325–1627 ppm (average 889 ppm). Hence, the REE content of the light-CL core is significantly lower than that of the dark-CL rim, and the difference between the two is mainly in HREE content. The light-CL core spots have Eu/Eu* of 0.07–0.60 (average 0.28) and Ce/Ce* of 1.89–24.27 (average 10.03). Because the contents of La and Pr are

typically present very low, Ce* in this study is obtained by the formulation $(Nd_N)^2/ Sm_N$
(Loader et al., 2017). The dark-CL rim spots have Eu/Eu* of 0.08–0.24 (average 0.13)
and Ce/Ce* of 6.57–200.31 (average 79.23). These results indicate that the light-CL
core have a weaker negative Eu anomaly and a weaker positive Ce anomaly than those
of the dark-CL rim. The inherited zircon spots have ΣREE of 602–1517 ppm, and show
depletion of LREE, enrichment of HREE, a positive Ce anomaly (Ce/Ce* of 1.52–
216.08) and a negative Eu anomaly (Eu/Eu* of 0.07–0.13) (Fig. 6c).

## 5.3 Zircon U–Pb and Hf isotope composition

Seventy-seven spots were analysed for U-Pb isotope composition from samples
XY-001 and XY-008. In the U-Pb Concordia diagram (Fig. 7a, c), both the light-CL
core and dark-CL rim spots overlap within uncertainty on the Concordia curve. There
is a large degree of overlap between the 29 spots of dark-CL rim and 32 spots of light-
CL core in terms of $^{206}Pb/^{238}U$ age although the average value for $^{206}Pb/^{238}U$ age is
higher in the 32 spots of light-CL core (Fig. 7e). On a single zircon, the $^{206}Pb/^{238}U$ age
of the light-CL core is older than that of the dark-CL rim (Fig. 5), but the two values
are within the error range of the in-situ LA-ICP-MS analyses (individual spot of ±3–5%
relative precision, Schmitz and Kuiper, 2013). In sample XY-001, 33 spots define a
weighted mean $^{206}Pb/^{238}U$ age of 160.7±1.1 Ma (2σ, MSWD=1.3; Fig. 7b). In sample
XY-008, 28 spots define a weighted mean $^{206}Pb/^{238}U$ age of 159.6±1.1 Ma (2σ,
MSWD=1.2; Fig. 7d). The other 10 spots with distinctly older ages ($^{207}Pb/^{206}Pb$ ages
ranging from 2500 to 2173 Ma) were obtained on inherited cores. Their ages are
discordant, suggesting that these inherited cores were variably influenced by lead loss.
Among these, 9 spots define a discordia line with an upper intercept age of 2163 ± 13
Ma (MSWD=0.45) (Fig. 7f).
Twenty-four zircons were analyzed for Lu-Hf isotope composition. The variation
in Hf isotopic data is limited, between 9 spots from light-CL core and 9 spots from dark-
CL rim. 18 spots exhibit a range of $^{176}$Hf/$^{177}$Hf ratios from 0.281921 to 0.282030, which
converts to εHf(t) values between -26.6 to -22.8 (Fig. 8), and two-stage Hf model ($T_{DM2}$)
ages of 2650 to 2889 Ma by using the U-Pb age for each zircon. Six analytical spots,
which define the Concordia upper intercept age of 2163 Ma, show $^{176}$Hf/$^{177}$Hf radios
and εHf(t) values of 0.281443 to 0.281496 and -0.7 to 1.5, respectively, with $T_{DM2}$ age
of 2648 Ma to 2791 Ma by using the upper intercept age.
**6. Discussion**
**6.1 Significance of the two stages of zircon**
Generally, zircon with high U content can easily break down into the metamict
state because of the radiation damage to the lattice caused by α-particles originating
from the decay of uranium (Mezger and Krogstad, 1997). The physical and structural
changes often lead to the loss of Pb and addition of trace elements such as LREE. In
this study, the dark-CL rim spots have high U content, which is significantly higher than
the median value of zircon U content in granitic magma (350 ppm, Wang et al., 2011).
Hence, the metamictization degree of the zircons must be taken into consideration. Data
from dark-CL rim spots plot on the Concordia curve, indicating no obvious Pb loss. The
internal structure of dark-CL rim is relatively intact, with obvious oscillatory zoning,
and few cracks, implying that the physical and structural of the dark-CL rim remained
unchanged. Nasdala et al. (1998) suggested that the metamictization of zircon can be
well characterized by Raman spectroscopy. The half-width of the $v_3$(SiO$_4$) Raman band
(*b*) of 10 cm$^{-1}$ and 20 cm$^{-1}$ are proposed to approximately distinguish well-crystallized,
intermediate and metamict zircons (Nasdala et al., 1998). The dark-CL rim have *b*
values of 5.4–9.2, characterizing them as well-crystallized. Therefore, the above
features indicate that the dark-CL rim are not metamict. Consequently, it can be
concluded that the U-Pb isotope and trace element systematics of the dark-CL rim have
not been changed by metamictization.
Both the light-CL core and dark-CL rim have oscillatory zoning patterns, and their
chondrite-normalized REE patterns are characterized by steeply positive slopes from
the LREE to HREE with strong negative Eu anomalies and pronounced positive Ce
anomalies. The above characteristics are consistent with those of igneous zircon
(Hoskin and Schaltegger, 2003). Although hydrothermal zircon can also have
oscillatory zoning patterns similar to magmatic zircons, there are obvious differences
in trace elements between the magmatic and hydrothermal zircon (Hoskin et al., 2005).
In the discrimination diagram (Fig. 9), both the spots of light-CL core and dark-CL rim
fall in or near the magmatic field, which is obviously different from hydrothermal
zircon. Hence, the above characteristics indicate that both the light-CL core and dark-
CL rim have a magmatic origin.
The light-CL core was overgrown continuously by the dark-CL rim. In addition,
the contact between the light-CL core and dark-CL rim is euhedral. Such core-mantle
overgrowth relationships indicate that the light-CL core domains are not inherited
zircons. The similar Hf isotopic data of the light-CL core and dark-CL rim is also
consistent with this interpretation. For the age population, the samples of XY-001 and
XY-008 have MSWD of 1.3 and 1.2, respectively, which are both within the expected
range for 95 % confidence interval (Mahon, 1996). Although the $^{206}Pb/^{238}U$ age of dark-
CL rim is generally younger than that of light-CL core, the ages of these two distinct
domains have the characteristics of continuous variation, and do not show two or more
distinct age populations (Fig. 7b, d). These phenomena do not support the presence of
antecrystic zircons (Siégel et al., 2018). Hence, both the light-CL core and dark-CL rim
are most likely autocrystic zircon formed in one distinct pulse of magma. The weighted
mean U-Pb ages of 160.7±1.1 Ma and 159.6±1.1 Ma can be interpreted as the
emplacement age of the Zhoujiapuzi granite. The obvious difference in internal
structure and trace element composition between the light-CL core and dark-CL rim
could be due to significant changes in their crystallization environments (Wang et al.,

2007).

The Zr/Hf ratio in zircon has a negative correlation with the degree of fractionation
in the parent melt (Claiborne et al., 2006). In this study, the Zr/Hf ratios of the dark-CL
rim (21–40) are obviously lower than those of the light-CL core (39–56) (Fig. 10a). In
addition, incompatible elements such as U and REE will become enriched in the highly
evolved magma (Zhao et al., 2014). In this study, the contents of U and REE of dark-
CL rim are significantly higher than those of light-CL core (Fig. 10a). Overall, the
above features reflect that the dark-CL rim crystallized from a later and more evolved
magma.
Watson and Harrison (2005) found that the Ti content of zircon has a strong
dependence on temperature (T), and obtained a Ti-in-zircon thermometer ($T_{Zr-Ti}$). Since
then, Ferry and Watson (2007) suggested that the solubility of Ti in zircon depends not
only on T and activity of $TiO_2$ ($aTiO_2$) but also on the activity of $SiO_2$ ($aSiO_2$), and
revised the $T_{Zr-Ti}$. We use the $T_{Zr-Ti}$ from Ferry and Watson (2007) and the recommended
values ($aSiO_2=1$, $aTiO_2 = 0.5$) for the activity of $SiO_2$ and $TiO_2$ (Schiller and Finger,
2019), due to the presence of ilmenite and quartz in the Zhoujiapuzi granite. The $T_{Zr-Ti}$
from the light-CL core and dark-CL rim are 684–830 °C (average 761 °C) and 509–
712°C (average 635 °C), respectively, i.e. the light-CL core formed at higher
temperatures than the dark-CL rim. The $T_{Zr-Ti}$ value shows a significant positive
correlation with Zr/Hf (a tracer of fractional crystallisation), and shows continual
fractionation and cooling (Fig. 10b). As the light-CL core and dark-CL rim formed in
different magmatic evolution stages, it is problematic to use the same $aSiO_2$ and $aTiO_2$
values to calculate both $T_{Zr-Ti}$ values for both. For ilmenite bearing granites, Schiller
and Finger (2019) suggested that the variation of $aTiO_2$ values corresponding to
different zircon crystallization stages is small. In addition, Schiller and Finger (2019)
showed that the $aSiO_2$ value of the ilmenite-bearing granites at the onset of magmatic
zircon crystallization was more than 0.75. Even if the $aSiO_2$ value of the light-CL core
is changed from 1.0 to 0.75, the temperature will only drop by about ~27 °C, which is
significantly lower than the 126 °C difference between the average $T_{Zr-Ti}$ value of the
light-CL core and dark-CL rim. Therefore, it is certain that the light-CL core formed at
higher temperatures than the dark-CL rim, although we cannot calculate the specific
temperature difference.

Cerium exists in magmas as both $Ce^{3+}$ and $Ce^{4+}$. Because the 0.84-Å radius of the

$Zr^{4+}$ ion is more closely matched by the $Ce^{4+}$ (0.97-Å radius) than the $Ce^{3+}$ (1.143-Å
radius) (all ionic radii are from Shannon, 1976), $Ce^{4+}$ is more compatible in the zircon
structure than the $Ce^{3+}$. Hence, the magnitude of Ce anomaly is a useful tool for
evaluating the oxygen fugacity condition of crystallization environment (e.g. Ballard et
al., 2002; Trail et al., 2012). Loader et al. (2017) suggested that the Ce/Ce* ratio is
likely to be the most robust measure of magma redox conditions, although it is only a
semi-quantitative measure. In this study, the Ce/Ce* ratio of the light-CL core and dark-
CL rim are 6.30–153.36 (average 32.51) and 21.81–5773.06 (average 787.39),
respectively. This result suggests that the dark-CL rim formed in a higher oxygen
fugacity environment than the light-CL core. As shown in the Ce/Ce*-Zr/Hf diagram
(Fig. 10c), Ce/Ce* has a significant negative correlation with Zr/Hf, showing that the
oxygen fugacity condition is increasing with the evolution of magma.
The absence of enclaves and disequilibrium textures in the Zhoujiapuzi granite
and uniform εHf(t) values of the light-CL core and dark-CL rim do not support magma
mixing and wall-rock assimilation. Consequently, the abrupt change between the
crystallization environment of the light-CL core and dark-CL rim is not due to the
magma mixing or contamination during magma evolution. Therefore, we propose that
the light-CL core was formed in a relatively deep magma chamber, which had low
oxygen fugacity, low Zr saturation and higher temperature. The low Th, U and REE,
and widely-spaced oscillatory zoning patterns indicate a low growth rate of zircon
(Hoskin and Schaltegger, 2003; Wang et al., 2011). In contrast, the dark-CL rim was
formed during the ascent and/or at the emplacement location of the magma. At this

stage, the oxygen fugacity significantly increased, the temperature decreased, and Zr saturation increased due to the crystallization differentiation. In this environment, the crystallization rate of zircon significantly increased, forming the zircons with a higher content of Th, U and REE elements, low CL emission and narrowly-spaced oscillatory zoning patterns.

Zircon U-Pb dating is the most commonly used method in geochronology, especially dating the emplacement age of magmatic rocks. A weighted mean age or upper intercept age is usually obtained to represent the emplacement time of a magmatic rock. However, the autocrystic zircons in this study record two different magmatic evolution stages. Previous studies, such as Wang et al. (2007), Zhao et al. (2014) and Chen et al. (2020), also show that zircons can crystallize continually or intermittently in a single phase of magmatism, showing several growth zones of clearly different internal structure and distinct time difference. Therefore, autocrystic zircon can be formed in two or more evolution stages during one distinct pulse or increment of magma. Some scholars even regard that the age difference of different stages can be more than dozens of Ma (Wang et al., 2007). Therefore, if the zircon ages in the same magmatic rock have a large range of variation, this could be caused by the zircons recording different stages in magmatic evolution, related to different levels of magma within the crust and/or different temperature regimes. In this paper, although the apparent age of the dark-CL rim is generally younger than that of the light-CL core, the age difference between the two is within the error range of the in-situ LA-ICP-MS analyses (individual spot of $\pm 3$–5% relative precision). Therefore, further work is

needed to verify the actual age difference between the two magmatic evolution stages.
Nevertheless, it is notable that the bulk petrology and geochemistry of the host pluton
does not record and reveal this two-stage magmatic evolution, which can only be
detected in the zircon analysis.

**6.2 Genetic type: I-type affinity**

The Zhoujiapuzi granite has low Zr (113﹣242 ppm), Ce (26.5﹣121.5 ppm),
Zr+Nb+Ce+Y (152.0﹣382.6 ppm), $(Na_2O + K_2O)/CaO$ (4.53﹣8.31) and $FeO^*/MgO$
(5.09﹣10.56), distinct from the typical A-type granites (Fig. 11a-d). Furthermore, the
Zhoujiapuzi granite does not contain mafic alkaline minerals, such as arfvedsonite,
riebeckite, etc., which is also distinctly inconsistent with typical A-type granites (Wu et
al., 2003). Wu et al. (2017) suggested that a high formation temperature is one of the
most important characteristics of A-type granite. Zircon saturation thermometry ($T_{Zrn}$)
and Ti-in-zircon thermometer ($T_{Zr-Ti}$) are two methods for estimating magma
temperatures. As noted above, because the values of $aSiO_2$ and $aTiO_2$ during the early
zircon crystallization cannot be accurately obtained, the temperature of this period
cannot be accurately obtained through the Ti-in-zircon thermometer. Zircon saturation
thermometry was introduced by Watson and Harrison (1983) and is suitable for non-
peralkaline crustal source rocks. Since the zircon solubility is mainly affected by
temperature, major element compositions have a limited impact on calculated $T_{Zrn}$
(Miller et al., 2003). In addition, the errors introduced by crystal-rich composition tend
to cancel as changes in Zr concentration and M value during crystallization have
opposite effects on the $T_{zrn}$ value (Miller et al., 2003). Therefore, the composition of
Zhoujiapuzi granite can be used to estimate the magma temperature. The calculated $T_{Zrn}$
values for the Zhoujiapuzi granite are in the range of 803-870 °C (mean=845 ±20°C).
It was proposed that the $T_{Zrn}$ suggests an upper limit on the temperature of melt
generation for inheritance-rich granitoid (Miller et al., 2003). Hence, the magma
temperature of the Zhoujiapuzi granite should be lower than or equal to the $T_{Zrn}$ value,
which is significantly lower than that of typical A-type granite (>900 ℃, Skjerlie and
Johnston, 1992; Douce, 1997). Thus, the Zhoujiapuzi granite is not an A-type granite.
The samples of the Zhoujiapuzi granite have A/KNC < 1.1, relatively high $Na_2O$
(3.96–4.65 wt.%) and lack peraluminous minerals (e.g. cordierite, andalusite,
muscovite and garnet), which are clearly different from S-type granites (Chappell and
White, 1992). With the rise of the degree of crystallization, $P_2O_5$ contents
(generally>0.1 wt.%) increase in S-type granites, accompanied by an
increase/immutability in $SiO_2$ (Wolf and London, 1994). However, the Zhoujiapuzi
granite samples have low $P_2O_5$ contents (0.02－0.08 wt.%), and decrease with
increasing $SiO_2$ (Fig. 11e), which are features consistent with the I-type granite rather
than S-type granite (Chappell and White, 1992). Additionally, Rb has a positive
correlation with Y (Fig. 11f), which has been considered as an indicator of I-type granite
(Jiang et al., 2018). Furthermore, the composition of the Zhoujiapuzi granite fall in the
I-type granite field in the discrimination diagrams of granites introduced by Collins et
al. (1982) (Fig. 11 c-d). Therefore, we conclude that the Zhoujiapuzi granite is a I-type
granite.

## 6.3 Petrogenesis of the high Sr/Y granite

The samples of the Zhoujiapuzi granite have high Sr/Y and $(La/Yb)_N$ ratios and low Y and Yb contents (Fig. 12a) consistent with the geochemical signatures of modern adakites (Defant and Drummond, 1990). However, other geochemical parameters of the Zhoujiapuzi granite, such as the high $K_2O/Na_2O$ ratio (0.93 –1.22), low $Al_2O_3$ content (14.49 –15.02%, except one) and Sr content (in half of the samples lower than 400 ppm), are obviously different from typical adakites ($K_2O/Na_2O \leqslant 0.42$, $Al_2O_3 \geqslant$ 15 %, Sr>400 ppm, Defant and Drummond., 1990; Drummond et al., 1996, Martin et al., 2005). A variety of petrogenetic models have been proposed for the origin of high Sr/Y magmatic rocks, such as partial melting of subducting oceanic crust (Model A, Defant and Drummond, 1990), delaminated lower continental crust (LCC) (Model B, Kay and Kay, 1993; Xu et al., 2002), differentiation of basaltic arc magma (Model C, Castillo et al., 1999), magma mixing between mantle-derived mafic and crust-derived silicic magmas (Model D, Ma et al., 2013a), partial melting of thickened basaltic LCC (Model E, Gao et al., 2004; Ou et al., 2017), or melting of a high Sr/Y (and La/Yb) source (Model F, Kamei et al., 2009; Ma et al., 2015).

### 6.3.1 Model A: Partial melting of subducting oceanic crust

The partial melting of the young, hot and hydrated subducted oceanic slab in the garnet stability field is the classical formation model of adakite (high Sr/Y rock) (Defant and Drummond, 1990). Studies have shown that the rock with this genetic model generally has the characteristics of high mantle components (such as MgO, CaO and Cr) because of the involvement of mantle magma (Wang et al., 2018). However, this phenomenon was not seen in the Zhoujiapuzi granite. In addition, the Zhoujiapuzi granite has high $K_2O/Na_2O$ ratios (0.92–1.22, average 1.13), which is inconsistent with

the slab-derived adakites ($K_2O/Na_2O$= ~0.4, Martin et al., 2005). Moreover, the low
εHf(t) values (-26.6 to -22.8) of the Zhoujiapuzi granite are also inconsistent with the
magmas derived from the partial melting of oceanic crust, which generally have
depleted isotopic character (Zhan et al., 2020). Furthermore, the Zhoujiapuzi granite
has low Ti/Eu and high Nd/Sm radios (Fig. 13a), and markedly negative Nb-Ta
anomalies (Fig. 4b), which are distinct from those of oceanic basalts (Yu et al., 2012).
In summary, the Zhoujiapuzi granite is difficult to explain by Model A.

### 6.3.2 Model B: Delaminated lower continental crust (LCC)

High-density, garnet-bearing mafic lower crust delaminating or foundering into
the asthenosphere mantle and subsequent interaction with mantle peridotite could
produce high Sr/Y magmas (Kay and Kay 1993). Because the melt formed by partial
melting of the delaminated lower crust would interact with mantle peridotite during
magma ascent, the high Sr/Y magmas related to this petrogenetic model generally have
high MgO, Mg# and $TiO_2$ (Gao et al., 2004; Ou et al., 2017; He et al., 2021). The MgO
(0.10– 0.44 wt.%), Mg# (15– 26) and $TiO_2$ (0.09– 0.34 wt.%) values of Zhoujiapuzi
granite are significantly lower than the above values (Fig. 13b- d). In addition,
delamination of the lower crust generally occurs in within-plate extensional settings
(Gao et al., 2004), and will form a large number of Mg-rich (Mg#>50) rocks due to the
partial melting of lithospheric mantle and/or upwelling of asthenosphere (Ou et al.,
2017). However, these Jurassic magmatic rocks in the Liaodong Peninsula are generally
considered to be formed in a compressional environment related to the subduction of
the Paleo-Pacific slab (Li et al., 2004; Yang et al., 2015a; Zhu and Xu, 2019; Zheng et
al., 2018). Furthermore, the middle-late Jurassic granites are generally Mg-poor (Fig.
13c). Due to the high temperature of the asthenosphere (1200 °C, Parsons and

McKenzie, 1978; King et al., 2015), rocks formed by partial melting of the delaminated lower crust should possess a high-temperature fingerprint. $T_{Zrn}$ has been used as a geothermometer to estimate partial melting temperatures (e.g., Miller et al., 2003; Collins et al., 2016). As mentioned before, the $T_{Zrn}$ of the Zhoujiapuzi granite is below 900 °C, which is markedly lower than the temperature of the asthenosphere. Therefore, the petrogenetic model of delaminated lower continental crust (Model B) is also inconsistent with the Zhoujiapuzi granite.

### 6.3.3 Model C: Differentiation of basaltic arc magma

Low-pressure fractional crystallization (involving olivine + clinopyroxene + plagioclase + amphibole+ titanomagnetite) or high-pressure fractional crystallization (involving garnet) from basaltic magmas have been proposed as two ways to generate adakitic characteristics (Castillo et al., 1999; Macpherson et al., 2006).

However, the composition of the Zhoujiapuzi granite is relatively uniform, including $SiO_2$, MgO and $Na_2O$, which does not support major fractional crystallization (Xue et a., 2017). Furthermore, the Zhoujiapuzi granite has abundant inherited zircons and no obvious depletion of Sr, Eu and Ba, showing that this granite has not experienced extensive fractionation (Miller et al., 2003). The samples form clear partial melting trends on the La/Yb versus La diagram (Fig. 13e), which also suggests that partial melting was more important than fractional crystallization (Gao et al., 2007; Shahbazi et al., 2021). In addition, crystal fractionation of basaltic melts can only form minor volumes of granitic melts, the ratio of the two is about 9:1 (Zeng et al., 2016). However, for the same age interval, no coexisting mafic-intermediate rocks have been found in the research area. In the wider region of the Liaodong Peninsula, Middle-Late Jurassic magmatism is dominated by felsic compositions; mafic- intermediate rocks are only

reported in the Huaziyu area (lamprophyre dikes, Jiang et al., 2005). Therefore, it is unlikely that there are large-scale mafic- intermediate rocks contemporaneous with the Zhoujiapuzi granite at depth according to the rock assemblage of Liaodong Peninsula in this period. Moreover, the zircon Hf isotopic compositions of the Zhoujiapuzi granite are quite different from those of the depleted mantle, but are similar to those of the basement (Liaohe Group and Liaoji granite) in the study area (Fig. 8). The ancient inherited zircons (2500 to 2173) with low $\varepsilon Hf(t)$ values also indicate older crustal material in the Zhoujiapuzi granite. For these reasons, it is highly improbable that Zhoujiapuzi granite was derived by differentiation of basaltic magma (Model C).

### 6.3.4 Model D: Magma mixing between mantle-derived mafic and crust-derived silicic magmas

The Zhoujiapuzi granite has high $K_2O/Na_2O$ ratio (>1) and A/CNK value (>1), together with the absence of mingling textures, mafic microgranular enclaves (MMEs), felsic xenocrysts and melting texture of plagioclase, implying that the mantle-derived magma is unlikely to have played an important role in the genesis of the Zhoujiapuzi granite (Castro et al., 1991). In addition, the Zhoujiapuzi granite is characterized by the development of biotite, but lacks amphibole and pyroxene. These features, coupled with the high A/CNK value, are consistent with an origin as a crust-derived granitoid, but obviously different from the granitoids formed by crust-mantle-derived magma mixing (Barbarin, 1990). Moreover, granites formed by magma mixing generally have high MgO, $TFe_2O_3$, CaO and Cr contents and low $SiO_2$ content (Ma et al., 2013a; Wang et al., 2018). These features are obviously inconsistent with the Zhoujiapuzi granite in this study. Additionally, the $\varepsilon Hf(t)$ values and trace element composition of the two stages of zircon also do not support magma mixing. Hence, magma mixing of mantle-derived

and crust-derived magmas (Model D) is also unlikely to have produced the Zhoujiapuzi
granite.

**6.3.5 Model E: Partial melting of thickened basaltic LCC**

Experimental studies have shown that the partial melt of basaltic LCC in the garnet
stabilization zone (> 40 km, i.e. ~1.2 GPa) can produce magma with a high Sr/Y ratio
(Rapp et al., 2003 and references therein). In these scenarios, high Sr/Y and overall
adakitic affinity are caused by leaving garnet as residual phases (e.g. Gao et al., 2004).
Based on geochemical data for the Zhoujiapuzi granites, partial melting of thickened
basaltic LCC is also unlikely to account for the high Sr/Y Zhoujiapuzi granite (Model
E). This conclusion is based on the following observations:
(1) This ratio of $(Gd/Yb)_N$ is the most important feature to judge whether garnet is
involvement in magma genesis (Ma et al., 2012). If the $(Gd/Yb)_N$ ratio of the source is
similar to the average value of the LCC (1.71, Rudnick and Gao, 2003), partial melting
of these crustal materials controlled by garnet at high pressure can produce melt with
$(Gd/Yb)_N$ of 5.8 (Huang and He, 2010). In contrast, the $(Gd/Yb)_N$ values (1.22–5.06,
average 2.69) of the Zhoujiapuzi granite are relatively low. (2) Studies of lower-crustal
xenoliths show that garnet may not be a common mineral in the lower crust of the NCC
(Ma et al., 2012). (3) As shown in the discrimination diagrams of granite sources (Fig.
13f, g), all samples fall in the range of metagreywacke-derived melts. Therefore, the
Zhoujiapuzi granite was considered to have been derived from crustal anatexis of
metagraywacke (or intermediate-acid igneous rock with similar mineral composition),
rather than basaltic lower crust.

**6.3.6 Model F: Melting of a high Sr/Y (and La/Yb) source**

Studies have shown that when a source rock has a high Sr/Y ratio, the high Sr/Y signature of the derived magma can inherit from their source, regardless of pressure (Kamei et al., 2009; Moyen, 2009; Ma et al., 2015). We suggest that partial melting of high Sr/Y Liaoji granite was most probably the origin of the high Sr/Y Zhoujiapuzi granite, as discussed below (Model F).

The Zhoujiapuzi granite has similar mineral assemblages (contains abundant K-feldspar and lacks hornblende) and geochemical composition (Fig. 13h) to the Tsutsugatake intrusion, which is explained by partial melting of arc-type tonalite or adakitic granodiorite (Kamei et al., 2009). Among the inherited zircons from Zhoujiapuzi granite, the $^{207}Pb$ / $^{206}Pb$ ages of all the spots are between 2132 and 2200 Ma, except one, and yield a Concordia upper intercept age of 2163 Ma. Both assimilation of country-rocks and incomplete melting of source rocks can explain the genesis of inherited zircon in granite. Due to the similar $T_{DM2}$ of autocrystic zircons (light-CL core and dark-CL rim) and inherited zircons, these inherited zircons most likely come from the source of the Zhoujiapuzi granite. In the study area, meta-sedimentary rocks and meta-volcanic rocks of the South Liaohe Group, Paleoproterozoic mafic rocks, as well as the Liaoji granites, have ~2.16 Ga zircon. In spite of an age peak of 2.17–2.16 Ga in detrital zircon age spectra of the metasediments from the South Liaohe Group, melting of a sediment-dominated source is unlikely to have occurred, as it would have also introduced other age peaks such as ~2.03 Ga and ~2.50 Ga (Li et al., 2015; Wang et al., 2020b). In addition, given the I-type characteristics of the Zhoujipuzi granite, derivation from an igneous precursor is more plausible rather than a metasedimentary origin (Chappell and White, 1992). Therefore, these ~2.16 Ga zircons from the Zhoujiapuzi granite are unlikely to come from the

South Liaohe Group. As shown in the host rock discrimination diagrams (Fig. 14,
introduced by Belousova et al., 2002), all the ~2.16 Ga inherited zircons from
Zhoujiapuzi granite fall into the granitoid area (Fig. 14), precluding that these ~2.16 Ga
zircon come from the Paleoproterozoic mafic rocks. In addition, the ~2.16 Ga inherited
zircons from Zhoujiapuzi granite and the zircons from the Liaoji granites lie in a similar
area in the $\varepsilon$Hf(t)-age (Ma) diagram (Fig. 8). Hence, the ~2.16 Ga inherited zircon most
likely come from the Liaoji granites.

Some of the Liaoji granites, such as the Muniuhe granite (comprising granodiorite

and syenogranite with no distinct boundary between the two), have adakitic signatures,
and similar REE and trace element patterns as the Zhoujiapuzi granite (Fig. 4). Based
on a model of batch melting (Shaw, 1970) using the experiments of Conrad et al. (1988),
the high Sr/Y characteristic of the Zhoujiapuzi granite can be explained by partial
melting of Muniuhe granitic pluton leaving amphibole as the main residue (Fig. 12b).

In our modelling, we choose the XY-005 sample to approximately represent the

primitive melt composition. The reasons are as below: as mentioned above, the high
Sr/Y characteristics of the Zhoujiapuzi granite are not caused by the fractional
crystallization of amphibole. Furthermore, the lack of positive correlation between
$Dy_N/Yb_N$ ratios and $La_N/Yb_N$ ratios (Fig. 13i) also suggests that fractional
crystallization of amphibole was not a significant process for the Zhoujiapuzi granite.
On the other hand, the samples of Zhoujiapuzi granite display variable Eu and Sr
contents, implying that plagioclase is likely a fractional phase. The separation of
titanomagnetite could explain the positive trend in $TFe_2O_3$ with increasing $TiO_2$ content,
consistent with the occurrences of magnetite in some studied rocks. This possible
mineral assemblage of fractional crystallization is also reflected by the chemical
variations in the Sr/Y-Y diagram (Fig. 12b). Hence, the sample XY-005, which has

highest Sr/Y, was chosen to represent a primitive melt composition. To find the best matching experimental melts, we have compared the major elements of the XY-005 sample with that of experimental melts and the characteristics of no garnet residue discussed above. Results are shown in Fig. 12b. The Sr and Y compositions of the starting material used in these experiments resemble those of the average composition of the Muniuhe granitic pluton (Sr=475 ppm, Y=9.77 ppm), if the residue contains a large volume of amphibole (>90 %). However, if more plagioclase is retained in the residue (e.g. 18.3 %), a source region with a higher Sr content is required. Therefore, a similar high Sr/Y Liaoji granite to the Muniuhe granitic pluton can produce the high Sr/Y signatures of the Zhoujiapuzi granite.

A large number of Yanshanian adakites (or high Sr/Y rocks) are developed in the NCC, which are generally considered to be derived from the thickened basaltic LCC (e.g. Gao et al., 2004; Wu et al., 2005a; Ma et al., 2013b). Zhang et al. (2001, 2003) suggested that these so-called "C-type adakites" indicated a large-scale crustal thickening event. However, according to the studies on the Triassic and Jurassic adakitic rocks near the Pingquan area, the northern part of the NCC, Ma et al. (2012, 2015) suggested that the adakitic signatures of these rocks are inherited from their source rocks. The research of the Zhoujiapuzi granite in this study also shows that among the widely distributed Jurassic high Sr/Y granites in the Liaodong Peninsula, there is at least one pluton with a high Sr/Y signature inherited from the source. Therefore, we suggest that melting of a high Sr/Y (and La/Yb) source is one of the important processes for the generation of Yanshanian high Sr/Y rocks in the NCC. This kind of high Sr/Y granite does not need to be formed in the garnet stability field.

**6.4 Tectonic implications**

A large number of Early Jurassic arc-like igneous rocks occur in the northeast part of NCC- Korean Peninsula-Hida belt, which belong to the middle-high K calc-alkaline series and are characterized by enrichment in LILE and depletions in HFSE (Wu et al., 2007; Tang et al., 2018 and references therein). In addition, the Early Jurassic accretionary complexes in the eastern margin of the Eurasian continent and the Japan islands, such as the Heilongjiang complex, the Khabarovsk complex and the Mino-Tamba complex, are considered to be related to subduction (Wu et al., 2007; Tang et al., 2018 and references therein). It is generally accepted that the Paleo-Pacific slab subducted westwards in the Early Jurassic (Tang et al., 2018; Zhu and Xu, 2018).

In the middle-late Jurassic, I-type granites are dominant in the Liaodong Peninsula, such as the Zhoujiapuzi granite (this study), Heigou pluton, Gaoliduntai pluton (Wu et al., 2005a), Waling granite (Yang et al., 2015a) and Wulong granite (Yang et al., 2018). There are not A-type granites, and mantle derived magmatism is extremely rare. These granites were formed by partial melting of crustal materials without obvious contribution of mantle derived magma (Wu et al., 2005a; Yang et al., 2015b, 2018; Xue et al., 2020). In addition, WNW-ESE compression during 157-143 Ma was widespread in the Liaodong Peninsula (Yang et al., 2004; Zhang et al., 2020). It not only mylonitized the granite plutons in middle-lower crust levels, but also intensely deformed the thick sedimentary cover in the upper crust (Qiu et al., 2018; Ren et al., 2020). Hence, Late Jurassic magmatism in the Liaodong peninsula is most likely to be related to subduction of the Paleo-Pacific plate in a mature continental arc, with crust

previously thickened by compressional tectonics, related to both the oceanic subduction and the earlier Mesozoic collisions at the north and south margins of the NCC. This setting would produce the conditions required for extensive crustal melting of pre-existing basement. There is a potential resemblance to the modern arc of the Central Andes (Allmendinger et al., 1997), where crustal thickening and plateau growth has developed over the Cenozoic (Scott et al., 2018), and melting of older basement has taken place during subduction of the Nazca plate (Miller and Harris, 1989). This model is also consistent with the idea that much of eastern China was a high orogenic plateau during the Mesozoic, before widespread Early Cretaceous extension and core complex development (Meng, 2003; Chu et al., 2020).

**7. Conclusions**

(1) LA-ICP-MS zircon U-Pb dating indicates that the Zhoujiapuzi granite in the Liaodong Peninsula formed at ~160 Ma.

(2) Zircon growth in Zhoujiapuzi granite can be divided into two distinct stages. The light-CL core was formed in a deeper, hotter, magma chamber, which had low oxygen fugacity and high temperature. The dark-CL rim formed from later, more evolved, magma. Oxygen fugacity significantly increased and the temperature decreased at this stage. The Zhoujiapuzi granite is a case study of multistage generation and emplacement of magma, revealed by zircons, where no signals are discernible in the bulk petrology or geochemistry.

(3) The I-type Zhoujiapuzi granite originated from partial melting of the

Paleoproterozoic Liaoji granites. The high Sr/Y compositions are inherited from their
source rocks, rather than being a direct indication of deep crustal melting, or any other
common mechanism for generating adakitic signatures.
(4) The Late Jurassic tectonic setting of the Liaodong Peninsula and the eastern
NCC resembled the modern orogenic plateau of the Central Andes, where silicic
magmatism may occur by partial melting of older continental crust in a compressional
environment, related to the subduction of the Paleo-Pacific plate.
**Acknowledgements**
We thank Dr. Wenzhou Xiao, Dr. Jiajie Chen and Dr. Quan Ou for constructive reviews
and useful suggestions. We are also grateful to Ying Liu, Chunying Guo, Jianxiong Hu,
Ziming Hu for their help with the field work. This research was funded by the National
Nature Science Foundation of China (Grants No. 42030809, 41772349, 41902075,
42002095; 42162013), China Scholarship Council (No. 202008360018), Geological
Exploration Program of China Nuclear Geology (Grant No. D1802), the research grants
from the East China University of Technology (Grant No. DHBK2017103), Open
Research Fund Program of State Key Laboratory of Nuclear Resources and
Environment (East China University of Technology) (Grant 2020NRE13).

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

**Table captions**
Table S1. Major element (wt. %) and trace element (ppm) compositions of the
Zhoujiapuzi granite
Table S2. Raman microprobe data
Table S3. The zircon major element (wt. %) and trace element (ppm) from the
Zhoujiapuzi granite
Table S4. Zircon La-ICP-MS U-Pb isotopic data and ages of the Zhoujiapuzi granite
Table S5. Zircon Hf isotopic data of the Zhoujiapuzi granite

**Figure captions**

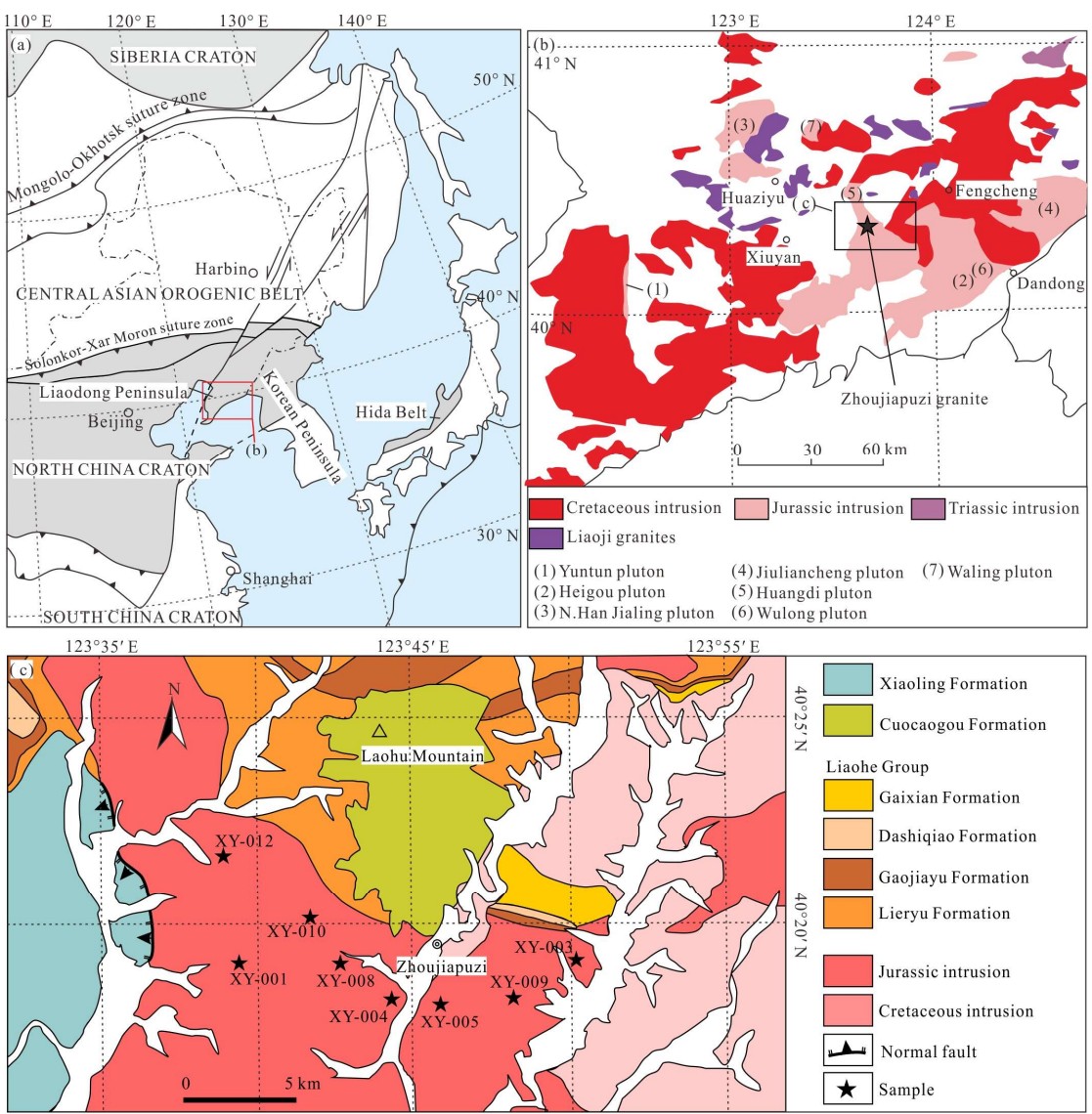


Figure 1. (a) Simplified geological map of Northeast China (Modified from Li et al.,
2016); (b) distribution of Mesozoic intrusions in the Liaodong Peninsula (Modified
from Wu et al., 2005a); (c) geological map of the Zhoujiapuzi granite.

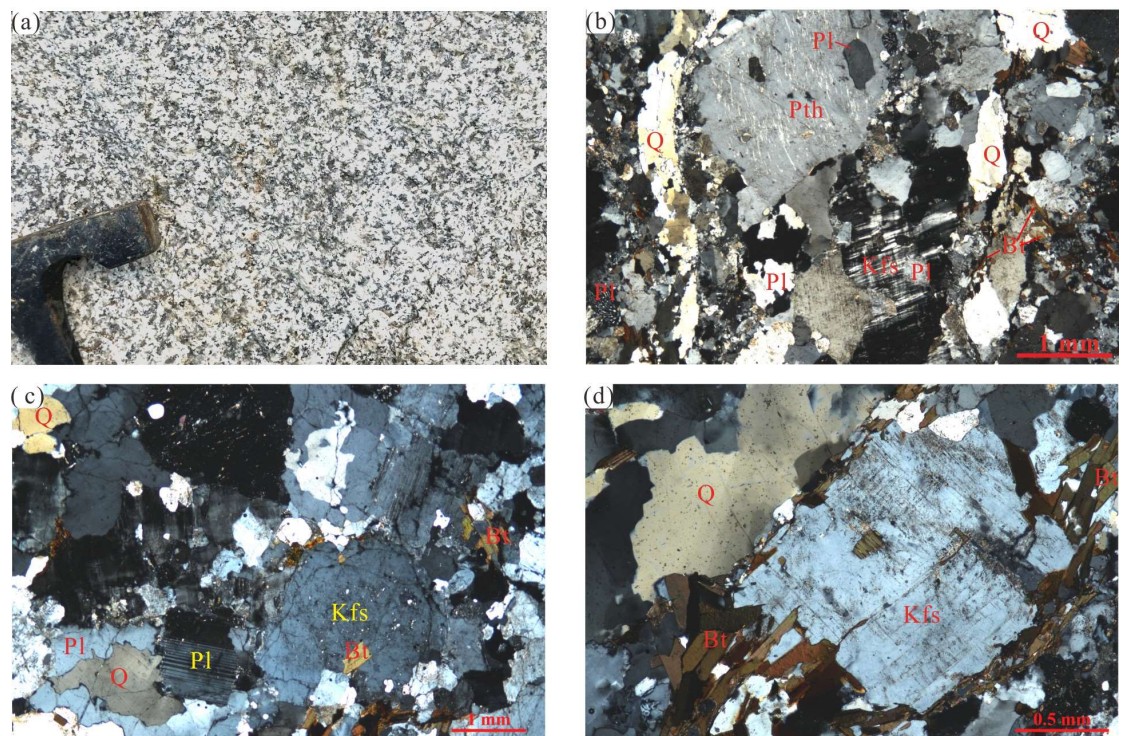


Figure 2. Outcrop photograph (a) and corresponding micrographs (b, c, d-
perpendicular polarized light). Q quartz; Kfs feldspar; Pl plagioclase; Pth perthite; Bt
biotite

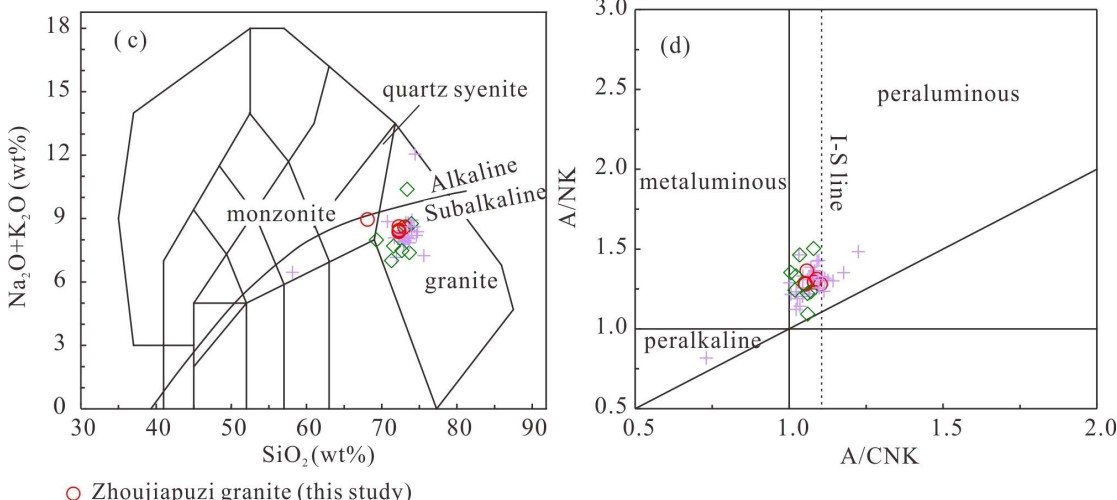

○ Zhoujiapuzi granite (this study)
+ Other middle-late Jurassic granitoids (165−156 Ma) in Liaodong Peninsula
 (Wu et al., 2005a; Yang et al., 2015b, 2018; Xue et al., 2020)
◇ Muniuhe granitic pluton (Yang et al., 2016; Wang et al., 2020a)


Figure 3. Geochemical classification diagrams for the Zhoujiapuzi granite. (a) TAS
diagram (after Frost et al.,2001); (b)A/CNK-A/NK diagram (after Maniar and
Piccoli, 1989)

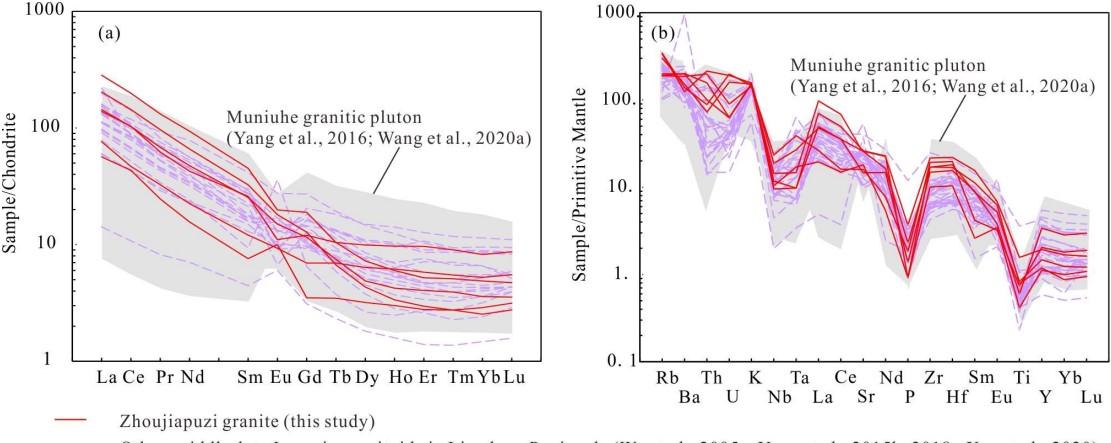


Figure 4. Chondrite-normalized REE patterns and primitive mantle-normalized trace
element patterns of the Zhoujiapuzi granite (chondrite and primitive mantle values are
from Sun and McDonough, 1989).

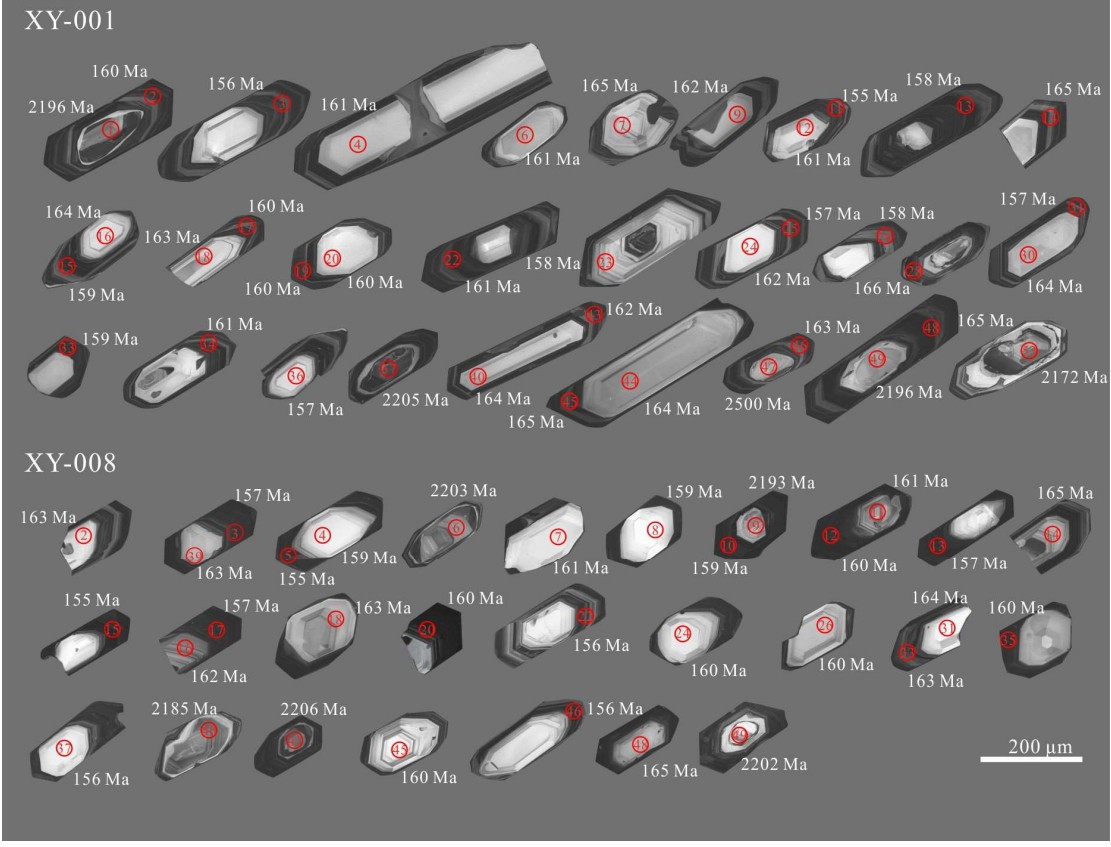


Figure 5. CL images of zircons. Circles denote U-Pb analysis spot. Numbers in the
circles are the spot numbers. Numbers near the analytical spots are the U–Pb ages
(Ma).

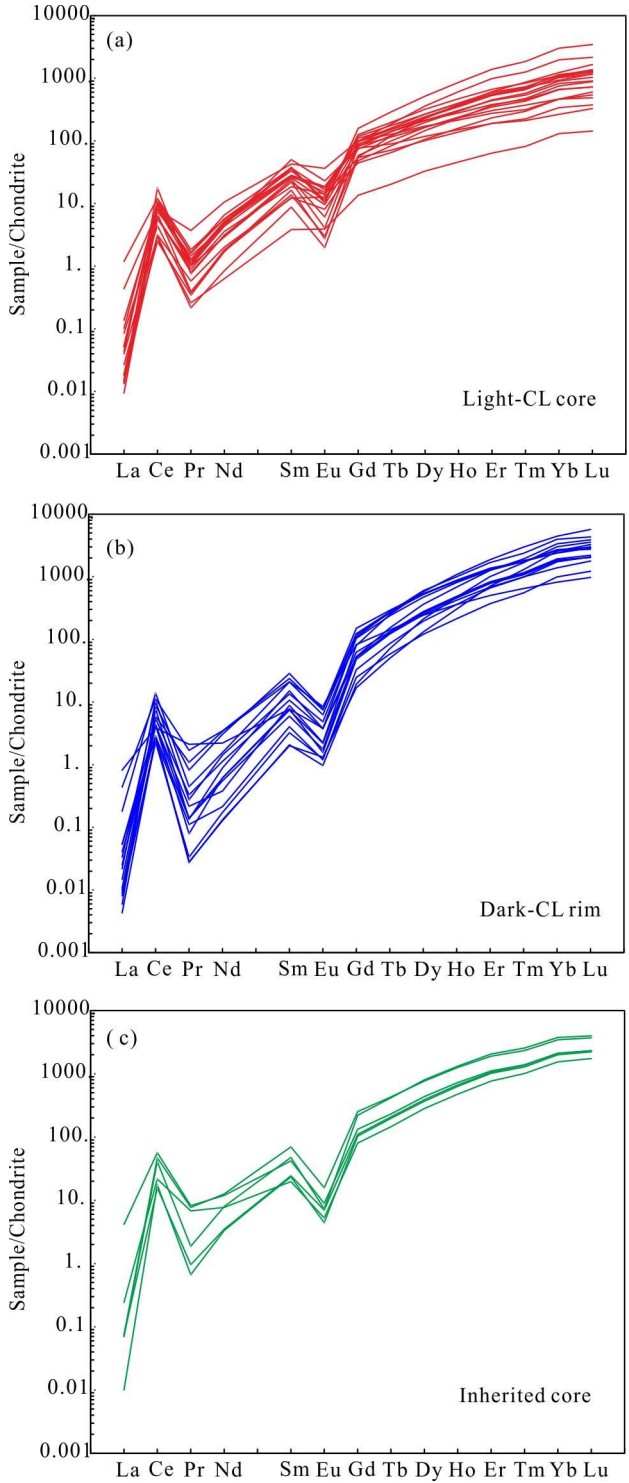


Figure 6. Chondrite-normalized REE patterns of zircon (chondrite values are from
Sun and McDonough, 1989).

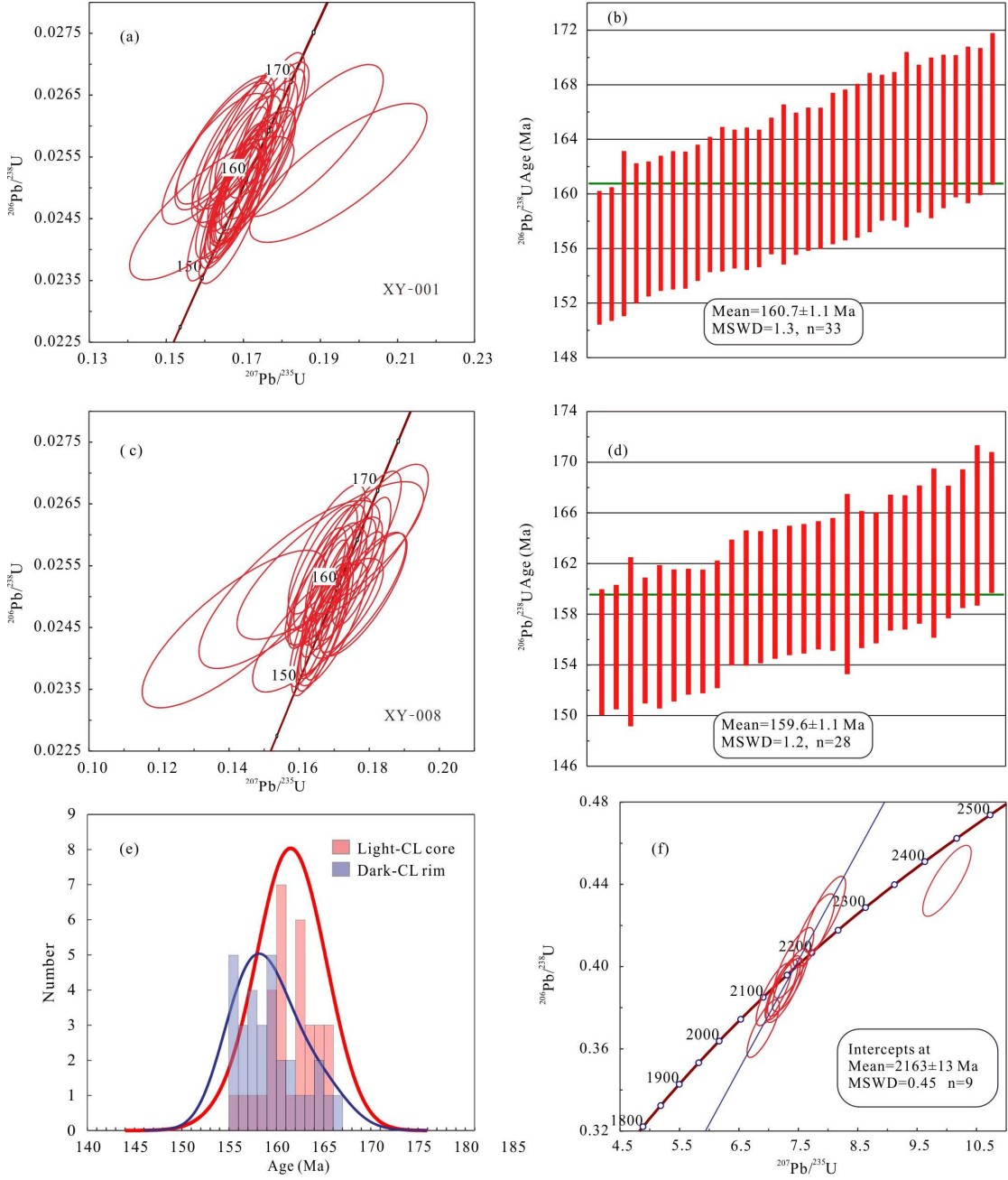


Figure 7. Concordia diagrams for zircon LA-ICP-MS U-Pb analyses.



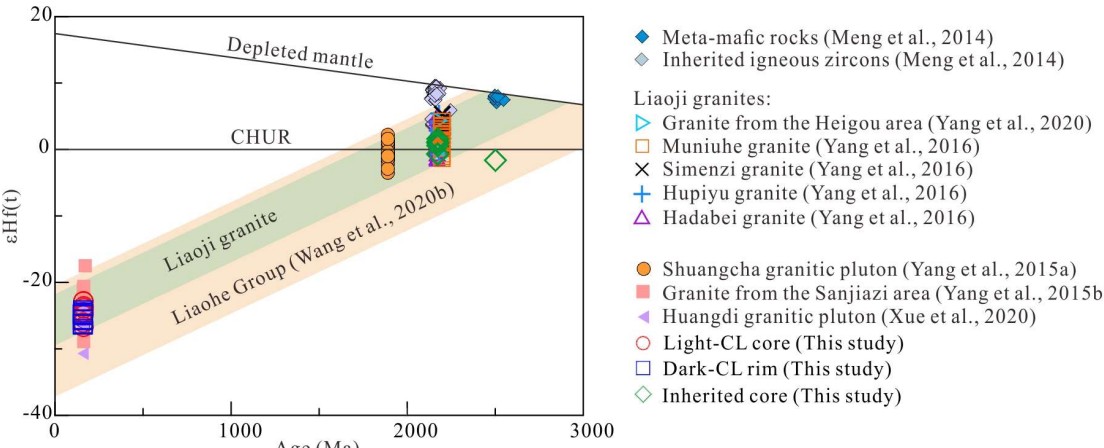

Figure 8. Zircon εHf(t)-age (Ma) diagram for samples in this study and published data for the region.

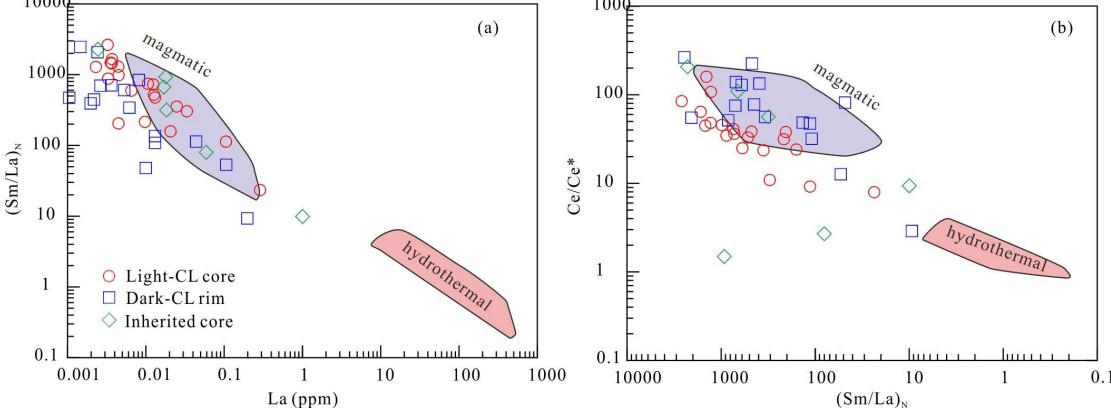

Fig. 9. Discrimination plots for magmatic and hydrothermal zircon (Hoskin, 2005).

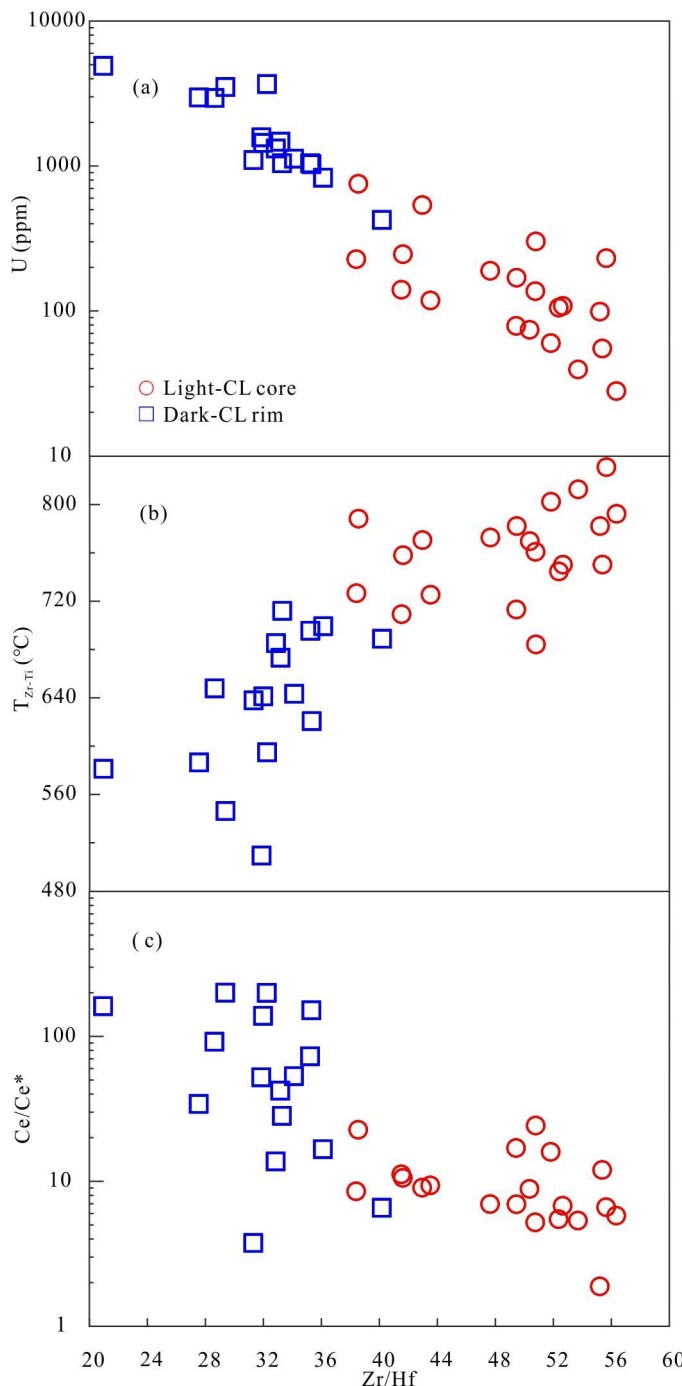


Figure 10. Covariation diagrams for zircon from the Zhoujiapuzi granite. (a) U vs.
Zr/Hf; (b) $T_{Zr-Ti}$ vs. Zr/Hf; (c) Ce/Ce* vs. Zr/Hf.

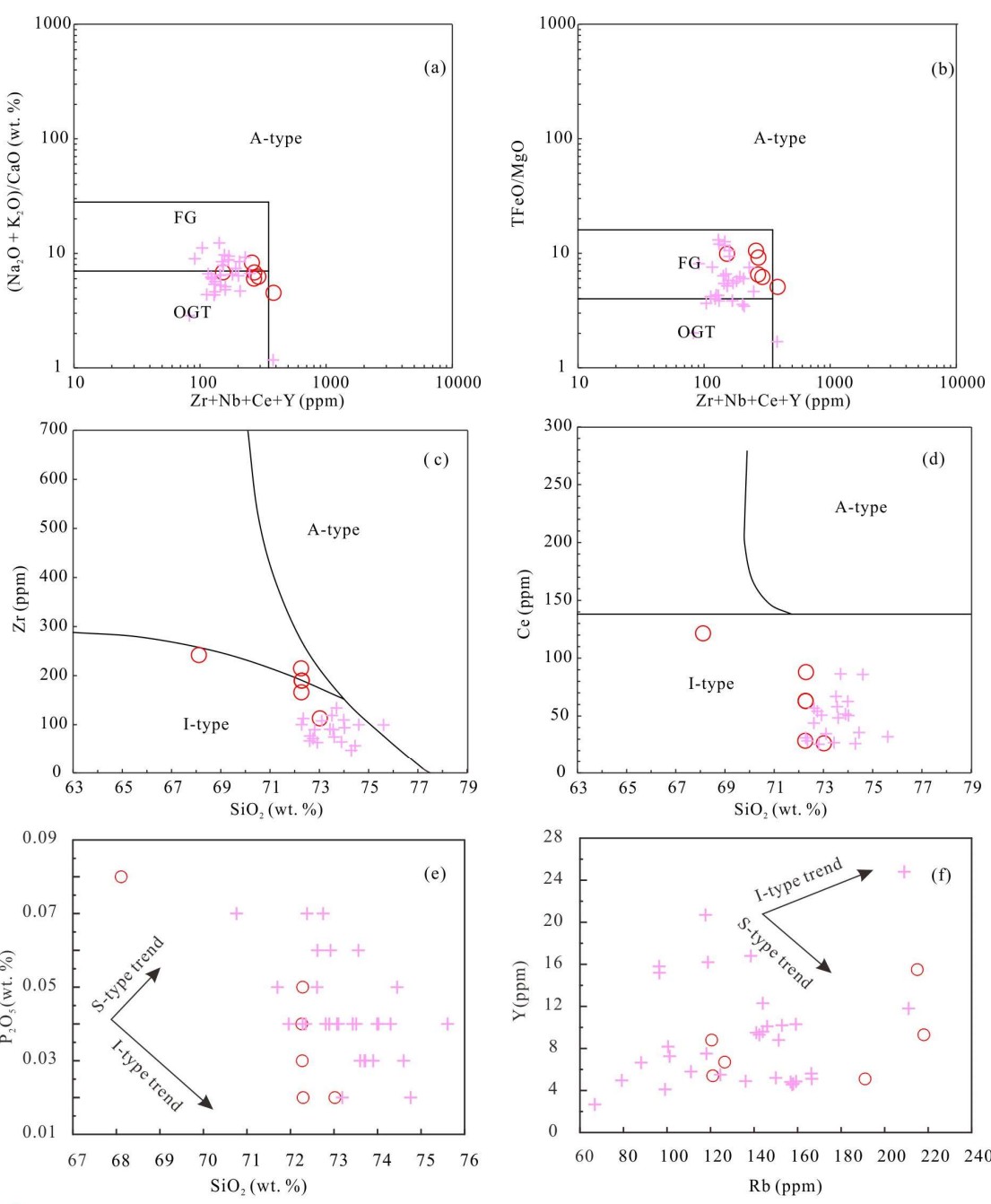

○ Zhoujiapuzi granite (this study)
+ Other middle-late Jurassic granitoids in Liaodong Peninsula (Wu et al., 2005a; Yang et al., 2015b, 2018; Xue et al., 2020)

Figure 11. Chemical variation diagrams for the Zhoujiapuzi granite. (a and b)

Zr+Nb+Ce+Y vs. $(Na_2O + K_2O)/CaO$ and TFeO/MgO (after Whalen et al., 1987); (c

and d) $SiO_2$ vs. Zr and Ce (after Collins et al., 1982); (e) $SiO_2$ vs. $P_2O_5$ diagram; (f)

Rb vs. Y diagram

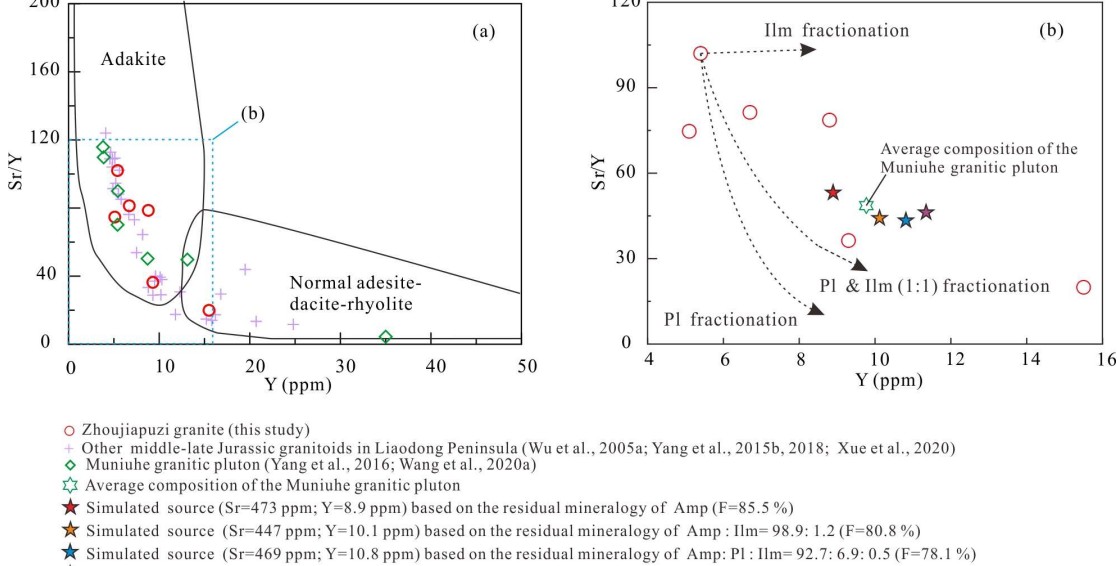

○ Zhoujiapuzi granite (this study)
+ Other middle-late Jurassic granitoids in Liaodong Peninsula (Wu et al., 2005a; Yang et al., 2015b, 2018; Xue et al., 2020)
◇ Muniuhe granitic pluton (Yang et al., 2016; Wang et al., 2020a)
☆ Average composition of the Muniuhe granitic pluton
★ Simulated source (Sr=473 ppm; Y=8.9 ppm) based on the residual mineralogy of Amp (F=85.5 %)
★ Simulated source (Sr=447 ppm; Y=10.1 ppm) based on the residual mineralogy of Amp : Ilm= 98.9: 1.2 (F=80.8 %)
★ Simulated source (Sr=469 ppm; Y=10.8 ppm) based on the residual mineralogy of Amp : Pl : Ilm= 92.7: 6.9: 0.5 (F=78.1 %)
★ Simulated source (Sr=524 ppm; Y=11.3 ppm) based on the residual mineralogy of Amp: Pl : Ilm= 82.4: 18.3 (F=72.2 %)

Figure 12. Adakite discrimination diagrams for the Zhoujiapuzi granite (after Defant and Drummond, 1990).

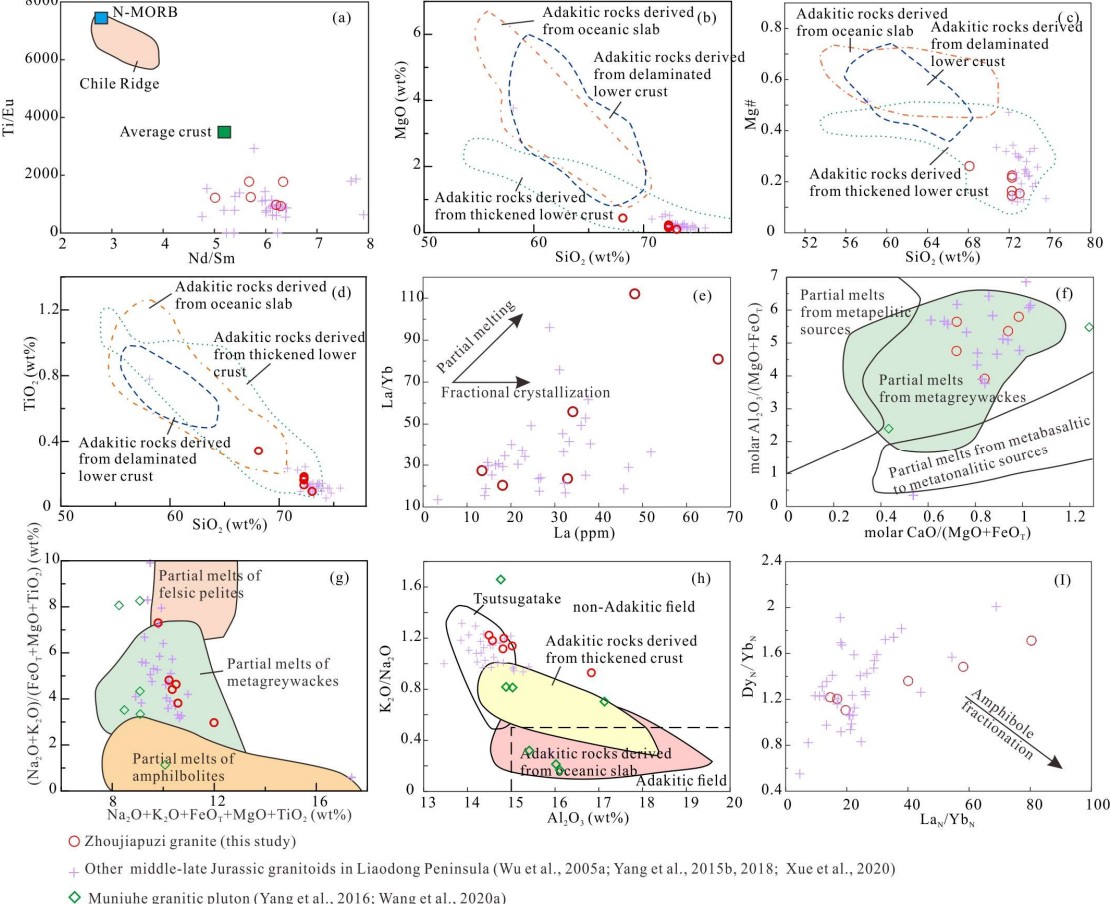

○ Zhoujiapuzi granite (this study)
+ Other middle-late Jurassic granitoids in Liaodong Peninsula (Wu et al., 2005a; Yang et al., 2015b, 2018; Xue et al., 2020)
◇ Muniuhe granitic pluton (Yang et al., 2016; Wang et al., 2020a)

Figure 13. Source characteristics (a-d and f-h) and crystal fractionation (e and i)

discrimination diagrams for the Zhoujiapuzi granite. Plots of (a) Nd/Sm vs. Ti/Eu (Yu
et al., 2012); (b-d) $SiO_2$ vs. MgO, Mg# and $TiO_2$ (after Wang et al., 2006); (e) La vs.
La/Yb (Gao et al., 2007); (f) molar $Al_2O_3$/(MgO+$FeO_T$) vs. molar CaO/(MgO+$FeO_T$)
(after Altherr et al., 2000); (g) ($Na_2O$+$K_2O$)/($FeO_T$+MgO+$TiO_2$) vs.
$Na_2O$+$K_2O$+$FeO_T$+MgO+$TiO_2$ (after Patiño Douce, 1999); (h) $K_2O$/$Na_2O$ vs. $Al_2O_3$
diagrams (after Kamei et al., 2009); (i) $La_N$/$Yb_N$ vs. $Dy_N$/$Yb_N$.

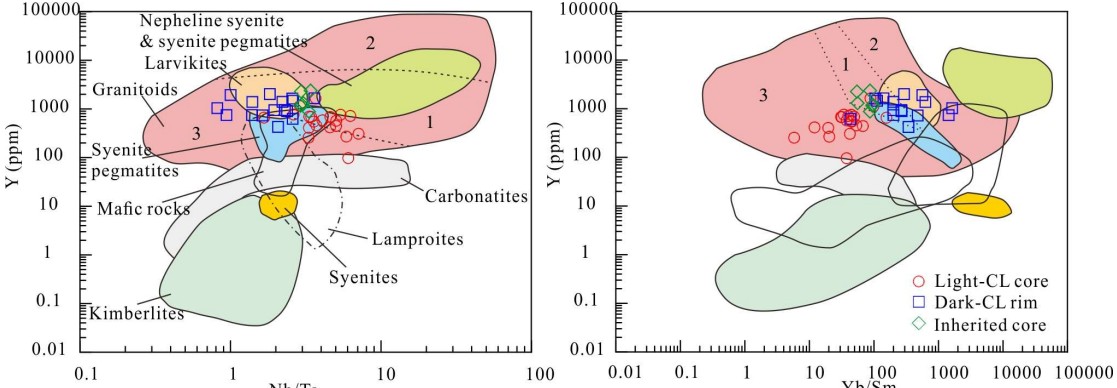


Figure 14. The fields of zircon compositions used as discriminants for different rock
types (after Belousova et al., 2002). 'Granitoids' include: 1 aplites and leucogranites;
2 granites; 3 granodiorites and tonalities
