# Peer review of "Whole-rock and zircon evidence for evolution of the Late Jurassic high Sr/Y Zhoujiapuzi granite, Liaodong Peninsula, North China Craton"

_Solid Earth, 2021_

## Author Comment (AC1)

Dear Reviewer:

Thank you for your comments concerning our manuscript entitled "Whole-rock and zircon evidence for evolution of the Late Jurassic high Sr/Y Zhoujiapuzi granite, Liaodong Peninsula, North China Craton" (ID SE-2021-129). Those comments are all valuable and very helpful for revising and improving our paper, as well as the important guiding significance to our researches. We have studied comments and suggestions carefully and have made correction. We hope meet with approval. Below the comments of the reviewer are response point by point and the revisions are indicated.

1) Response to comment: I am not convinced by the geochronological data being able to discriminate between two different zircon populations. The geochronology part would benefit from a more careful data treatment. Sometimes individual dates are presented without uncertainties even when trying to resolve a difference of a few Myr (~1-2%), which is below the resolution of individual data-points. should not be the case. I am not familiar with the approach of only illustrating 1s uncertainties as in most studies 2s is used. This would further highlight the overlapping nature of all analysed Jurassic zircons. The weighted means have very low uncertainties (<1%), which to me is maybe a slight overinterpretation of the data. Horstwood et al., 2016 is a good reference for robust data treatment for geochronological data. I would think that with a more conservative approach the temporal differences would fade away. Also, I am surprised that apparently all cores are older than the rims even though they typically fall within analytical uncertainty. I have rarely seen this in an in-situ data-set. Having said that as far as I can tell the data looks high quality. However, it is essential to report standard analyses and their reproducibility. I am sure the authors just forgot to add it but without those standard data the analytical data are not useable. The same applies for the zircon trace element data and Hf isotope data.

Thank you for your valuable comment. We used 2S uncertainties to reprocess the original data. In the process of data processing, the background of 1-10 s and the integral interval of 20-55s are used for both standard spots and measuring spots. We

adopted a more conservative method to process the zircon data, and put the data of light CL core and dark CL rim together to calculate the weighted average age. The ages of 160.7 ± 1.1 Ma (MSWD=1.3) and 159.6 ± 1.1 Ma (MSDW=1.2) were obtained for the two samples. The MSDW are both within the expected range for 95 % confidence interval. In this revised MS, the age data, trace element data and Hf isotope data of the standard zircons in the analyses are added in section 4 and Supplementary data.

2) Response to comment: The exclusion of other possible explanations appears slightly simplistic at times. An example would be referring to the small amount of silicic melt generated from differentiation of mafic magma but not addressing the same point about the partial melting of meta-granitoids (The Jurassic intrusions are much more exposed than the Lioaji granites). I believe going into more detail in the individual discussion points would help. Also, comparing the data to other studies and describing similarities or differences might help. If more than one model cannot be fully excluded it is fair to say that as well. Not every data-set can identify the one and only solution.

We gratefully appreciate your valuable suggestion. In the process of revision, we read a large number of papers and added relevant proofs and diagrams in the part excluding other possible explanations, such as models A, B and C. The details will be explained in more detail in the following answers.

We are sorry that some contents are not clearly stated in the MS. The Liaoji granite is more extensive than what is shown in Fig. 1 and lies within an area measuring 300 km × 70 km. We have made a supplementary description in section "3. Samples and petrography". In addition, the late Jurassic granite shown in Figure 1 is actually composed of several different granite bodies, but their boundaries are indistinguishable and connected together. Hence, this paper just suggests that the granite near the Zhoujiapuzi area is most likely been formed by the partial melting of Liaoji granite. We have modified the text in the MS:

*"The research of the Zhoujiapuzi granite in this study also shows that among the widely distributed Jurassic high Sr/Y granites in the Liaodong Peninsula, there is at least one pluton with a high Sr/Y signature inherited from the source."*

In addition, based on a model of batch melting, the Zhoujiapuzi granite is formed by partial melting of Liaoji granite with a high degree (>80 %). Hence, the volume of the Zhoujiapuzi granite and Liaoji granite will not be much different. Therefore, regarding the size of the Zhoujiapuzi granite, it is not a surprise that there is Liaoji granite of corresponding size.

As you mentioned, we cannot completely rule out other possibilities, but model 6 is obviously more reasonable.

3) Response to comment: I struggle with the current temperature/fO2 discussion. It might benefit from being a bit more in-depth in the main manuscript. See Loader et al., 2022; Loucks et al., Schiller and Finger and others.

We gratefully appreciate your valuable suggestion and references. We have read the paper, such as Schiller and finger, 2019, Loader et al., 2017, etc. Since FeO and Fe2O3 are not distinguished in this study, it is impossible to obtain aTiO2 and aSiO2 in different periods through rhyolite-MELTS or Perple_X. However, through the comparison of aTiO2 and aSiO2 between the early melt and the late melt in the extreme condition, we can get the relative temperature during zircon crystallization at the core and edge.

The same reasoning applies for oxygen fugacity. Since this paper only needs to obtain the relative relationship of oxygen fugacity between early melt and late melt semi-quantitatively, we use the Ce/Ce* value recommended by Loader et al., 2017 for comparison. Because the contents of La and Pr are typically present at very low, Ce* in this study is obtained by the formulation $(Nd_N)^2/ Sm_N$.

4) Response to comment: Line 22: The abstract changes rapidly from being descriptive to the conclusion part. Just saying "Interpretation of the elemental and isotopic data suggests" does not illustrate how the conclusions were derived. It would be good if a few lines were added highlighting how the conclusions were derived.

We have rewritten the abstract and added the process of reaching a conclusion.

5) Response to comment: Line 38: It would help the reader if the reason for this interpretation by most authors was briefly explained.

We appreciate the suggestion. In the second paragraph of the introduction, we briefly explain the views of different authors.

6) Response to comment: Line 41: In granitic rocks in general or those in the NCC?

The research by Kamei et al., 2009 is in Japan; the research by Ma et al., 2015 is in NCC; the research by Zhan et al., 2020 is in the Qinghai-Tibet Plateau. The above researches show that crustal thickening is not a necessary condition for the generation of all high Sr/Y rock pluton deriving from crust source.

7) Response to comment: 128-132: The relatively broard interpretation here comes very early in the manuscript purely based on CL images. Some Zircons illustrate CL patterns of a dark core surrounded by a lighter domain and again a dark rim. This is not discussed in the text so far. Are the dark cores considered to be inherited or do these zoning systematics suggest a more dynamic system than just early and late crystallisation?

Considering the Reviewer's suggestion, we have changed the "early stage of zircon" and "late stage of zircon" to "light-CL core" and "dark-Cl rim", respectively. In addition, we then explained the possible zircon areas in the light CL cores, which are inherited zircons.

*"According to the CL images, most zircons show an internal division into 2 distinct domains: light-CL core and dark-CL rim. The light-CL core is characterized by bright CL intensity and widely-spaced oscillatory zoning patterns. The dark-CL rim is overgrown continuously by the light-CL core and is characterized by extremely low CL emission and narrowly-spaced oscillatory zoning patterns. In addition, some zircons have inherited cores, which have corroded and rounded shapes in contact with the light-CL core, such as 1# and 37# in XY-001 and 6# and 41# in XY-008 (Fig. 5). These inherited zircons have oscillatory zoning in CL images."*

8) Response to comment: 152: I would say the dates overlap within uncertainty.

The two stages of zircon age are indeed indistinguishable in the test method of this research. In the revised MS, we combined the age data of "light CL core" and "dark CL rim" to calculate the weighted average age and obtained a reasonable MSWD value.

9) Response to comment: 153: I couldn't find the grain with supposedly inverse zoning of the U-Pb dates.

Sorry for the inaccuracy description. We have rewritten this part.

*"In the U-Pb Concordia diagram (Fig. 7a, c), both the light-CL core and dark-CL rim spots overlap within uncertainty on the Concordia curve. There is a large degree of overlap between the dark-CL rim and light-CL core in terms of $^{206}Pb/^{238}U$ age although the mean value for $^{206}Pb/^{238}U$ age is higher in the light-CL core (Fig. 7e). On a single zircon, the $^{206}Pb/^{238}U$ age of the light-CL core is older than that of the dark-CL rim (Fig. 5)."*

10) Response to comment: 166: The assumptions to determine TDM2 need to be described in more detail.

Thanks for your comments, in the original version we described it at the end of table S5. However, this is easily overlooked. Therefore, in the revised MS, we describe it in section "4. Analytical methods".

11) Response to comment: 189: What is the definition here of "same magmatism"? Magma reservoir, plumbing system, trans crustal mush, magma chamber? There are very different models about the architecture of magmatic systems, it would be good to be precise here. Especially, as different models use different assumptions on the potential timescales of the magmatic systems.

Thank you for your opinion. "One distinct pulse of magma" (Miller et al., 2007) is a more accurate expression of our meaning than "same magmatism".

12) Response to comment: 190: And what about antecrysts (Miller et al., 2007)? Considering the Reviewer's suggestion, we have added a part to judge whether it is antecrystic zircon or autocrystic zircon:

*"For the age population, the samples of XY-001 and XY-008 have MSWD of 1.3 and 1.2, respectively, which are both within the expected range for 95 % confidence interval (Mahon, 1996). Although the $206Pb/238U$ age of dark-CL rim is generally older than that of light-CL core, the ages of these 2 distinct domains have the characteristics of continuous variation, and do not show 2 or more distinct age populations (Fig. 7b, d). These phenomena do not support the presence of antecrystic zircons (Siégel et al., 2018). Hence, both the light-CL core and dark-CL rim are most likely autocrystic zircon formed in one distinct pulse of magma."*

13) Response to comment: 196: What would be the reference to use those activities for that mineral assemblage? It needs to be argued why it I valid to use the same activities

for both zircon generations. Schiller and Finger could be a good reference here. Also Gualda and Ghiorso. They also highlight the variation of aTiO2 within individual systems. It might be worth to propagate that uncertainty onto the uncertainty of the temperatures.

We appreciate for the suggestions and references. We have paid attention to the influence of magma composition evolution on the Ti thermometer. Since FeO and Fe2O3 are not distinguished in this study, it is impossible to obtain aTiO2 and aSiO2 in different periods through rhyolite-MELTS or Perple_X. However, through the comparison of aTiO2 and aSiO2 between the early melt and the late melt in the extreme condition, we can get the relative temperature of zircon crystallization at the core and edge. We also explain the temperature data in the MS:

*"It is worth noting that the light-CL core and dark-CL rim are formed in different magmatic evolution stages. Hence, using the same aSiO2 and aTiO2 values to calculate the TZr-Ti value of both light-CL core and dark-CL rim is problematic.*

*.....*

*Therefore, it is certain that the light-CL core formed at higher temperatures than the dark-CL rim, although we can't get the specific temperature difference."*

14) Response to comment: 198: Could a zircon crystallisation temperature of 498C maybe suggest that not all calculated data are valid?

We have realized that the current whole rock data cannot be directly used to calculate aTiO2 and aSiO2 at various stages of magma evolution. Therefore, in this MS, we emphasize the relative temperature relationship between the early stage and late stage, rather than absolute temperature value. Individual spots have obviously low value, which may be related to test error, or hitting small inclusions.

15) Response to comment: 199: Please explain the relevance of the correlation between U and Ti.

This is a valuable comment. We have changed the abscissa element from U to Zr/Hf, because this value directly represents the evolution degree of magma (as described above in the MS).

In addition, we have added a sentence to explain this relevance:

*"The $T_{Zr-Ti}$ value shows a significant negative correlation with Zr/Hf (a tracer of fractional crystallisation), and shows continual fractionation and cooling (Fig. 10b)"*

16) Response to comment: 200ff: Maybe see Loader et al., 2022 for a thorough description different potential sources for the Ce anomaly.

Sorry, we didn't find the paper of loader et al., 2022, but we found the paper of loader et al., 2017 in EPSL. According to his suggestion, we used Ce/Ce* to judge the oxygen fugacity. Although it is a semi quantitative method, it meets the need of comparing the relative relationship between early stage and late stage oxygen fugacity in this study.

17) Response to comment: 204: Again the role of U needs to be discussed in more detail. It is always referred to but the petrogenetic reasons for it are not explained.

Considering the Reviewer's suggestion, we uniformly use Zr/Hf as the abscissa in Fig. 10 and explain Zr/Hf as "a tracer of fractional crystallization".

18) Response to comment: 205: I would rather say it suggests. Imply is a very strong wording, which I personally would not be comfortable with in this case.

We have changed "imply" to "suggest".

19) Response to comment: 207: I doubt that Breiter is the original reference for this. Claiborne et al., 2006 might be better.

Thank you for pointing this out. We have revised this reference

20) Response to comment: 220 ff. This assumes that the magmatic history was very simple over long timescales: just cooling and differentiation over at least 3 Myr with a change in locus in between. Maybe refer to studies that suggest something similar. At the moment most studies suggest far more complex and dynamic magmatic systems.

Thank you for your valuable comment. The magmatic history of Zhoujiapuzi granite in this study is relatively simple. With the magma evolution, the melt temperature decreases and the oxygen fugacity increases, no obvious magma injection is found. Zircons are only autocrystic zircons and inherited zircons, with no antecrystic zircon. Hence, it is impossible to determine whether there are multiple partial melting events. The zircons in the rim and core are also basically within the error range. Whether this pulse of magmatic activity operated for a long time needs further work. Therefore, it seems unwise to build a more complex model based on the existing evidence.

21) Response to comment: 227: Magmatic rock dating = geochronology?

Thank you for pointing this out. We have rewritten this sentence:

*"Zircon U-Pb dating is the most commonly used method in geochronology, especially dating the emplacement age of magmatic rocks"*

22) Response to comment: 228: Only in this example it appears to be two stages. It could be more and it could be different for any other magmatic system.

We apologize for the inaccuracy of our formulation, and we have rewritten this paragraph.

*"However, the autocrystic zircons in this study record two different magmatic evolution stages. Previous studies, such as Wang et al. (2007), Zhao et al. (2014) and Chen et al. (2020), also show that zircons can crystallize continually or intermittently in a single phase of magmatism, showing several growth zones of clearly different internal structure and distinct time difference. Therefore, autocrystic zircon can be formed in two or more evolution stages during one distinct pulse or increment of magma."*

23) Response to comment: 226-237: In situ geochronology typically really struggles to resolve different magmatic events within single magmatic systems. Especially, at the age range investigated in this study. Ii would suggest referring to CA-ID-TIMS work here to make a more robust point.

As you said, the LA-ICP-MS method really cannot distinguish the ages of these two stages, and we have also explained it in this MS:

*"In this paper, although the apparent age of the dark-CL rim is generally older than that of the light-CL core, the age difference between the two is within the error range of the in-situ LA-ICP-MS analyses (individual spot of ±3–5% relative precision, Schmitz and Kuiper, 2013). Therefore, further work is needed to verify the actual age difference between the two magmatic evolution stages."*

24) Response to comment: 254-256: This assumes that the WR geochemistry is equivalent to the melt chemistry the first zircon crystallised from. It would be good to outline why this assumption is valid.

Thank you for your valuable comment. We have quoted the contents of Miller et al. (2003) on geology for explanation.

*"Zircon saturation thermometry was introduced by Watson and Harrison (1983) and is suitable for non-peralkaline crustal source rocks. Since the zircon solubility is mainly*

*affected by temperature, major element compositions have a limited impact on calculated TZrn (Miller et al., 2003). In addition, the errors introduced by crystal-rich composition tend to cancel as changes in Zr concentration and M value during crystallization have opposite effects on the Tzrn value (Miller et al., 2003). Therefore, the composition of Zhoujiapuzi granite can be used to estimate the magma temperature."*

25) Response to comment: 256-257: I can not follow this point. Please clarify. A zircon crystallisation temperature does not automatically mean that the zircon dissolves immediately at that temperature. Also, what is an initial temperature? It should also be addressed that typically zircon is considered to crystallise late during magma evolution.

Thank you for your comments. We have rewritten this section and deleted the expression "initial temperature".

*"The calculated TZrn values for the Zhoujiapuzi granite are in the range of 803-870 °C (mean=845 ±20°C). It was proposed that the TZrn suggests an upper limit on the temperature of melt generation for inheritance-rich granitoid (Miller et al., 2003). Hence, the magma temperature of the Zhoujiapuzi granite should be lower than or equal to the TZrn value, which is significantly lower than that of typical A-type granite (>900 °C, Skjerlie and Johnston, 1992; Douce, 1997)."*

26) Response to comment: 288 ff: As mentioned above the temperature argument is currently not very strong. Zircons typically do not record the high-temperature magmatic stage.

We appreciate for the suggestions. In this MS, we added other evidences and diagrams to prove that the Zhoujiapuzi granite does not belong to A-type granite. Therefore, the evidence of temperature is only one of many lines of evidence.

In addition, TZrn has also been used as a geothermometer to interpret the peak temperatures that magmatic rocks experienced, and to estimate partial melting temperatures (e.g., Miller et al., 2003; Collins et al., 2016; Siegel et al., 2018). We have also explained the validity of the Tzrn in this revised MS. If this is not reasonable, we can delete the section on temperature, which will not affect the integrity of this paper.

**References**

Miller, C. F., McDowell, S. M., Mapes, R. W., 2003. Hot and cold granites? Implications of zircon saturation temperatures and preservation of inheritance. Geology 31, 529-532.

Collins, W.J., Huang, H.-Q., Jiang, X., 2016. Water-fluxed crustal melting produces Cordilleran batholiths. Geology 44 (2), 143–146.

Siégel, C., Bryan, S. E., Allen, C. M., Gust, D. A., 2018. Use and abuse of zircon-based thermometers: A critical review and a recommended approach to identify antecrystic zircons. Earth-Sci Rev 176, 87-116.

27) Response to comment: 295 – 316: I find the argumentation against differentiation of basaltic magma slightly selective. SiO2 is not the greatest proxy for melt differentiation if only granites are exposed. The resolved SiO2 window is very small. Maybe the variation seen in the granites is just a matter of slightly different accumulation of minerals in different parts of the pluton? Also, major and trace elements do not necessary show similar signatures (see Klaver et al., 2017). The last point about the volumes is typically dealt with by mafic magmas composing le lower arc crust while more differentiated ones migrated upwards. I think a more thorough comparison with other studies might be beneficial in really discarding this model. Especially, work by Jagoutz and others on the Kohistan arc.

According to your opinion, we have rewritten this part. The change of SiO2 is really too small, which affects the persuasion of crystallization differentiation. Therefore, we have deleted the description of crystallization differentiation in this section.

The coexisting mafic-intermediate rocks may indeed be deep, but according to the late Jurassic magmatic rock assemblage of the Liaodong Peninsula, this possibility is relatively low. Of course, this evidence alone cannot exclude the model. We added other evidence in the revised MS:

*"However, the composition of the Zhoujiapuzi granite is relatively uniform, including $SiO_2$, MgO and $Na_2O$, which does not support major fractional crystallization (Xue et a., 2017). Furthermore, the Zhoujiapuzi granite has abundant inherited zircons and no obvious depletion of Sr, Eu and Ba, showing that this granite has not experienced extensive fractionation (Miller et al., 2003). The samples form clear partial melting trends on the La/Yb versus La diagram (Fig. 13e), which also suggests that partial melting was more important than fractional crystallization (Gao et al., 2007; Shahbazi et al., 2021). In addition, crystal fractionation of basaltic melts can only form minor volumes of granitic melts, the ratio of the two is about 9:1 (Zeng et al., 2016). However, for the same age interval, no coexisting mafic-intermediate rocks have been found in the research area. In the wider region of the Liaodong Peninsula, Middle-Late Jurassic magmatism is dominated by felsic compositions; mafic- intermediate rocks are only reported in the Huaziyu area (lamprophyre dikes, Jiang et al., 2005). Therefore, it is unlikely that there are large-scale mafic- intermediate rocks contemporaneous with the Zhoujiapuzi granite at depth according to the rock assemblage of Liaodong Peninsula in this period. Moreover, the zircon Hf isotopic compositions of the Zhoujiapuzi granite are quite different from those of the depleted mantle, but are similar to those of the basement (Liaohe Group and Liaoji granite) in the study area (Fig. 8). The ancient inherited zircons (2500 to 2173 Ma) with low εHf(t) values also indicate older crustal material in the Zhoujiapuzi granite. For these reasons, it is highly improbable that Zhoujiapuzi granite was derived by differentiation of basaltic magma (Model C)."*

28) Response to comment: 346-347: Not being familiar with the study of Kamei it is unclear for what reason they argue for a partial melting origin. Can the same line of

According to Kamei's research, there is a type of high Sr/Y granite with different mineral and geochemical characteristics and different genetic types from typical adakite, and this type of rock is named "pseudo adakites" by him. The judgment process is similar to that in this paper. It also excludes other possibilities and then obtains the possibility that the source area is arc-type tonalite or adakitic grandiorite through simulation.

The similar mineral assemblages and geochemical composition between the Zhoujiapuzi granite and Tsutsugatake intrusion are just one of the lines of evidence that the Zhoujiapuzi granite is derived from the partial melting of the Liaoji granite. We moved this part to the second section. In the original version, it seems that this evidence confirms that the high Sr/Y characteristics of Zhoujiapuzi granite are inherited from the source area.

29) Response to comment: 347-348: Previously it was argued that only very little felsic melt can be generated by the differentiation of basaltic melt. But isn't the same true for the partial melting of a granite? In fig. 1 the Liaoji granites are far less exposed than the Jurassic granites. How would this be reconciled in this model?

We are sorry that some contents are not clearly stated in the MS. Liaoji granite is more extensive than what is shown in Fig. 1 and lies within an area measuring 300 km × 70 km. They have tectonic contact mostly with the Liaohe Group, but locally they occur as the base of the Liaohe Group or as intrusions therein (Liu et al., 2018). We have made a supplementary description in the section "3. Samples and petrography".

In addition, the late Jurassic granite shown in Figure 1 is actually composed of several different granite bodies, but their boundaries are indistinguishable and gradational. Hence, this paper just suggests that the granite near the Zhoujiapuzi area is most likely been formed by the partial melting of Liaoji granite. I'm sorry we didn't make it clear in the original MS. We have modified the expression in the revised MS. Moreover,

based on a model of batch melting, Zhoujiapuzi granite is formed by partial melting of Liaoji granite with a high degree (>80 %). Hence, the volume of the Zhoujiapuzi granite and Liaoji granite will not be much different, which is different from the ~ 9:1 ratio between basic rocks and acid rocks in the model of differentiation.

30) Response to comment: 366-368: The diagram 13b) seems to illustrate fractionation paths. Wouldn't this rather favour a differentiation origin?

We are sorry that we have not clearly described the crystallization differentiation of this granite. Crystal differentiation of plagioclase and Ilm does exist in the evolution of Zhoujiapuzi granite. However, it is not the main factor controlling the composition of the granite. We added the following in section 6.3.3 to explore the degree of crystalline differentiation

*"However, the composition of the Zhoujiapuzi granite is relatively uniform, including SiO2, MgO and Na2O, which does not support major fractional crystallization (Xue et a., 2017). Furthermore, the Zhoujiapuzi granite has abundant inherited zircons and no obvious depletion of Sr, Eu and Ba, showing that this granite has not experienced extensive fractionation (Miller et al., 2003). The samples form clear partial melting trends on the La/Yb versus La diagram (Fig. 13e), which also suggests that partial melting was more important than fractional crystallization (Gao et al., 2007; Shahbazi et al., 2021)."*

31) Response to comment: 368-369: Alternatively, crystallisation of amphibole would result in the opposite fractionation path. The presence of plagioclase does not imply that it was the main fractionation phase as the in-situ plagioclase has no fractionation effect on the WR-chemistry.

Thank you for your comments. We are sorry for not giving a full explanation here. In the revised MS, we explained why it is mainly affected by the crystallization differentiation of plagioclase:

*"In our modelling, we choose the XY-005 sample to approximately represent the primitive melt composition. The reasons are as below: as mentioned above, the high Sr/Y characteristics of the Zhoujiapuzi granite are not caused by the fractional crystallization of amphibole. Furthermore, no positive correlations between DyN/YbN ratios and LaN/YbN ratios (Fig. 13i) also suggest that fractional crystallization of amphibole was not a significant process for the Zhoujiapuzi granite. On the other hand, the samples of Zhoujiapuzi granite displayed variable Eu and Sr contents, implying that plagioclase is likely a fractional phase. The separation of titanomagnetite could explain the positive in TFe2O3 with increasing TiO2 content, consistent with the occurrences of magnetite in some studied rocks. This possible mineral assemblage of fractional crystallization is also reflected by the chemical variations in the Sr/Y-Y diagram (Fig. 12b). Hence, the sample XY-005, which has highest Sr/Y, was chosen to represent a primitive melt composition."*

32) Response to comment: 410: It reads like they granite crystallised over 4 Myr. I am sceptical that the data can resolve this. Also, did all zircons crystallise in-situ at the emplacement level? Previously, and in the second point it is argued that the early zircons might have formed deeper in the crust.

We adopted a more conservative method to process the zircon data, and put the data of light CL core and dark CL rim together to calculate the weighted average age. The ages of 160.7 ± 1.1 Ma (MSWD=1.3) and 159.6 ± 1.1 Ma (MSDW=1.2) were obtained for the two samples. The MSDW are both within the expected range for 95 % confidence interval.

33) Response to comment: 412ff: I struggle with the current temperature/fO2 discussion. It might benefit from being a bit more in-depth in the main manuscript. bulk-rock geochemistry per definition gives only one value. It would be surprising to resolve multiple stages from it.

This is a valuable comment. We have read the papers, such as Schiller and finger, 2019, Loader et al., 2017, etc. Since FeO and Fe2O3 are not distinguished in this study, it is impossible to obtain aTiO2 and aSiO2 in different periods through rhyolite-MELTS or Perple_X. However, through the comparison of aTiO2 and aSiO2 between the early melt and the late melt in the extreme condition, we can get the relative temperature of zircon crystallization at the core and edge.

The same for oxygen fugacity. Since this paper only needs to obtain the relative relationship of oxygen fugacity between early melt and late melt semi quantitatively, we use the Ce/Ce* value recommended by Loader et al., 2017 for comparison. Because the contents of La and Pr are typically present very low, Ce* in this study is obtained by the formulation $(Nd_N)^2/ Sm_N$.

34) Response to comment: Fig 4. What is the reason for the Tm-anomaly in some of the literature data? I can't think of many petrological reasons and it might be the result of sample contamination? Potentially by a flux if the data was generated from XRF beads. In any case, if there is no explanation for it might be better to discard that data.
The data with Tm anomaly is quoted from Yang et al., 2015b, 2018 (the same person). The test method of trace elements is to prepare samples by acid dissolution method and test them with HR-ICPMS (Element I) inductively coupled plasma mass spectrometer. The Tm anomaly phenomenon is not explained in his paper. In consideration of your opinion, we deleted this group of data in the REE patterns diagram.

35) Response to comment: Fig. 11d) Were all zircons analysed in the same counting mode? Systematic uncertainties could arise from analysing zircon with hugely varying U contents. It would be good to exclude this and describe how the data was analysed and whether a correction was applied.
In the section of "4. Analytical methods", we explained the analytical method:

*"The ICP-MS detector has dual-modes: pulse for lower signal, and analog for higher signal. Pulse-analog cross calibration was performed before the measurement of U-Pb isotopes, delivering a wider linear dynamic range – up to 10 orders of magnitude. For a signal of 238U higher than 1.2–1.4 Mio cps, equivalent zircon contains U concentrations higher than 600 ppm, and are measured in analog mode."*

36) Response to comment: Fig. 12: The range in SiO2 is very limited and the arrows often seem to rely on the single slightly lower SiO2 value. Is there maybe any more mafic data that could be illustrated?

Thank you for your valuable comment. There is only one late Jurassic basic rock reported in the Liaodong Peninsula, and it has no genetic connection with Zhoujiapuzi granite, so it is not added. As you mentioned, the variation range of SiO2 is small. Therefore, we deleted the two figures of SiO2 vs. TiO2 and SiO2 vs. TFeO. Because there are a large number of other evidence, the deletion of the above categories will not affect the description and interpretation.

37) Response to comment: Fig. 13b) the average composition of the Munihue granite does not seem to fall into the centre of the 6 clustering individual data points. I assume it requires the data point at ~35 ppm Y, which massively pulls the average to higher Y contents. It would be good to argue why the high Y data point is not an outlier. Especially, as it also falls outside the range of other Jurassic granites.

Thank you for your valuable comment.

The Muniuhe pluton is a highly fractionated granite, which is composed of two types of rock, granodiorite and syenogranite, without boundary. Hence, the variation range of REE and trace elements in Muniuhe pluton is relatively large. The sample with Y=~35ppm has the most significant negative Eu anomaly and low SiO2, Al2O3, CaO, and Na2O, indicating that it may have experienced strong plagioclase crystallization differentiation. The high content of REE, especially Ce, Y, Th and La, may be caused

by the accumulation of allanite. Therefore, the special geochemical composition of this sample may be due to the crystallization differentiation. In addition, The Liaoji granites are composed of a large number of granitic plutons, and their geochemical properties are quite different. For example, the ~2.17 Ga Hadabei granite has similar trace elements composition to this sample (Y=~35 ppm). Therefore, Liaoji granite with this chemical composition does exist. Therefore, the above phenomenon shows that the Liaoji granite in the source area of the Zhoujiapuzi granite may be composed of a variety of rocks with large differences in composition.

In fact, if this point with abnormally low Sr/Y is deleted, the higher Sr/Y source can form a pluton with the geochemical properties of the Zhoujiapuzi granite without experiencing a high degree of partial melting, which is easier.

38) Response to comment: Fig. 14a) The preferred interpretation in this manuscript is partial melting of a granitic source. This figure does not support this point very much. Thank you for your comments. We have replaced fig.14a with the more commonly used discriminant graph (La/Yb vs. La). The La/Yb vs. La diagram suggests that partial melting was more important than fractional crystallization.

We tried our best to improve the manuscript. We appreciate for Reviewer's warm work earnestly, and hope that the correction will meet with approval.
Once again, thank you very much for your comments and suggestions.

Sincerely,
Renyu Zeng

---

## Author Comment (AC2)

Dear Reviewer:

Thank you for your comments concerning our manuscript entitled "Whole-rock and zircon evidence for evolution of the Late Jurassic high Sr/Y Zhoujiapuzi granite, Liaodong Peninsula, North China Craton" (ID SE-2021-129). Those comments are all valuable and very helpful for revising and improving our paper, as well as the important guiding significance to our researches. We have studied comments and suggestions carefully and have made correction. We hope meet with approval. Below the comments of the reviewer are response point by point and the revisions are indicated.

To Reviewer #1:

1) Response to comment: The major criticism to the manuscript is about the tectonic implications, which is confusing and seems to have weak connection with the conclusion of this manuscript. The tectonic setting of the Late Jurassic granites given by the authors is unclear. In line 402-403, it is suggested that the Late Jurassic magmatism in Liaodong is related to the thinning of the NCC mantle lithosphere, which means an extensional setting since the thinning of lithosphere often occurs in thus setting. Whereas, in line 421, the authors give a compressional environment for those Late Jurassic granites, which is opposite to the previous statement. Besides, the authors proposed a mature continental arc setting for the Late Jurassic rocks, which I guess might be one of the implications of this work for the tectonic evolution of the NCC. However, the arguments for this implication are not well given and more discussion is needed.

We are very sorry for this error. In fact, the content of line 402-403 is what we should have deleted in the final version of MS. But we missed it. Therefore, we have deleted this content.

We gratefully appreciate your valuable suggestion. We have added more text and new references on this point. The revised content is as follows:

*"In the middle-late Jurassic, I-type granites are dominant in the Liaodong Peninsula, such as the Zhoujiapuzi granite (this study), Heigou pluton, Gaoliduntai pluton (Wu et*

*al., 2005a), Waling granite (Yang et al., 2015a) and Wulong granite (Yang et al., 2018). There are not A-type granites, and mantle derived magmatism is extremely rare. These granites were formed by partial melting of crustal materials without obvious contribution of mantle derived magma (Wu et al., 2005a; Yang et al., 2015b, 2018; Xue et al., 2020). In addition, WNW-ESE compression during 157-143 Ma was widespread in the Liaodong Peninsula (Yang et al., 2004; Zhang et al., 2020). It not only mylonitized the granite plutons in middle-lower crust levels, but also intensely deformed the thick sedimentary cover in the upper crust (Qiu et al., 2018; Ren et al., 2020). Hence, Late Jurassic magmatism in the Liaodong peninsula is most likely to be related to subduction of the Paleo-Pacific plate in a mature continental arc, with crust previously thickened by compressional tectonics, related to both the oceanic subduction and the earlier Mesozoic collisions at the north and south margins of the NCC. This setting would produce the conditions required for extensive crustal melting of pre-existing basement. There is a potential resemblance to the modern arc of the Central Andes (Allmendinger et al., 1997), where crustal thickening and plateau growth has developed over the Cenozoic (Scott et al., 2018), and melting of older basement has taken place during subduction of the Nazca plate (Miller and Harris, 1989). This model is also consistent with the idea that much of eastern China was a high orogenic plateau during the Mesozoic, before widespread Early Cretaceous extension and core complex development (Meng, 2003; Chu et al., 2020)."*

2) Response to comment: The 206Pb/238U ages for the ESZ and LSZ are undistinguished within the analytical error. The authors are not suggested to use these age data to discuss the different crystallization stages for the zoned zircons. In line 222-226, the dispersion of age data for zircon grains from the same sample are used to indicate the cooling rate of magma. What is the rationale? How to build the connection between the U-Pb isotopic variation to the cooling rate of magma? Please give more discussion about this linkage.

We agree with this comment. It is really true that we can not use the age data to discuss the different crystallization stages for the zoned zircons. We adopted a more

conservative method to process the zircon data, and put the data of light CL core and dark CL rim together to calculate the weighted average age. The ages of $160.7 \pm 1.1$ Ma (MSWD=1.3) and $159.6 \pm 1.1$ Ma (MSDW=1.2) were obtained for the two samples. The MSDW are both within the expected range for 95 % confidence interval. In addition, the dispersion of age data can not be used to indicate the cooling rate of magma. Therefore, we deleted that part (line 222-226) and deleted Fig.11d.

3) Response to comment: Except for the Liaohe Group, a lot of Precambrian granitic intrusions and mafic dikes/sills were also exposed in the Liaodong Peninsula, which are suggested to be included in the section of geological setting.

Thank you for your comment. In the section of geological setting, we supplement the overview of Precambrian magmatism. The details are as follows:

*"The study area experienced strong magmatic activity in the Paleoproterozoic, which can be divided into two stages of 2.2–2.1 Ga and ~ 1.85 Ga. The 2.18–2.14 Ga Liaoji granites (also called gneissic granites), which lie within an area measuring 300 km × 70 km, are dominated by A- and I-type granites (Li and Zhao, 2007; Yang et al., 2016; Wang et al., 2020a). Metamorphosed volcanic rocks (leptynite, leptite and granulite) in the Liaohe Group also formed at 2.2–2.1 Ga (Li et al., 2015). The ~1.85 Ga granites mainly consist of I- and S-type porphyry granites and alkaline syenites (Yang et al., 2007; Yang et al., 2015b). In addition, there were small amounts of mafic magmatic activity at ~2.17 Ga, ~2.1 Ga and ~1.8 Ga (Meng et al., 2014; Yuan et al., 2015). There are a variety of viewpoints on the Paleoproterozoic tectono-magmatic evolution in the Liaodong Peninsula, such as an intracontinental rift opening-closing model (Li et al., 2005) and an arc-continent collision model (Faure et al., 2004)."*

4) Response to comment: Please give the standard reference materials used in the dating and Hf isotope analyzing and their analytical results, which is important for readers to evaluate the data quality.

Considering the Reviewer's suggestion, we have supplemented the standard reference materials used in the dating and Hf isotope analysis and their analytical results in section "4. Analytical methods" and Supplementary data.

5) Response to comment: Line 170: please add the range of U concentration for high-U zircons as well as the cited reference.

Thank you for pointing this out. We checked the relevant literature, and there is no quantitative standard for the U content of high U zircon (e.g. Mezger et al., 1997; Zhao et al., 2014; Park et al., 2016). Considering that the median value of zircon U content in granite is 350 ppm (Wang et al., 2011), the description of "the LSZ have high U content" should be reasonable. Hence, considering the preciseness of the MS, the sentence "*the LSZ are characterized by high U content*" was replaced by "*the LSZ have high U content, which is significantly higher than the median value of zircon U content in granitic magma (350 ppm, Wang et al., 2011)*"

**References**

Mezger, K., Krogstad, E. J., 1997. Interpretation of discordant U-Pb zircon ages: An evaluation. J Metamorph Geol 15, 127-140.

Zhao, K., Jiang, S., Ling, H., Palmer, M. R., 2014. Reliability of LA-ICP-MS U-Pb dating of zircons with high U concentrations: A case study from the U-bearing Douzhashan Granite in South China. Chem Geol 389, 110-121.

Park, C., Song, Y., Chung, D., Kang, I., Khulganakhuu, C., Yi, K., 2016. Recrystallization and hydrothermal growth of high U–Th zircon in the Weondong deposit, Korea: Record of post-magmatic alteration. Lithos 260, 268-285.

Xiang, W., Griffin, W. L., Jie, C., Pinyun, H., Xiang, L. I., 2011. U and Th contents and Th/U ratios of zircon in felsic and mafic magmatic rocks: Improved zircon‑melt distribution coefficients. Acta Geologica Sinica‑English Edition 85, 164-174.

Wang X, Griffin WL, Chen J, Huang PY, Li X., 2011. U and Th contents and Th/U ratios of zircon in felsic and mafic magmatic rocks: improved zircon-melt distribution coefficients. Acta Geologica Sinica-English Edition 85, 164–74.

6) Response to comment: the citation should be 'Yang et al., 2015a' and the intrusion should be 'Wulong granite'. Sanguliu granites were formed in early Cretaceous.

We are very sorry for this mistake. We have rechecked the references of this MS.

7) Response to comment: The depiction for some figures is too simple, like Fig 13, 14, which is a bit odd.

Thank you for your comment. We have modified the captions of Fig. 13, Fig. 14, as well as Fig. 3 and Fig. 11. The details are as follows:

*"Figure 3. Geochemical classification diagrams for the Zhoujiapuzi granite. (a) TAS diagram (after Frost et al.,2001); (b)A/CNK-A/NK diagram (after Maniar and Piccoli, 1989);*

*Figure 11. Covariation diagrams for zircon from the Zhoujiapuzi granite. (a) U vs. Zr/Hf; (b) TZr-Ti vs. Zr/Hf; (c) Ce/Ce\* vs. Zr/Hf.;*
*Figure 13. Adakite discrimination diagrams for the Zhoujiapuzi granite (after Defant and Drummond, 1990);*

*Figure 14. Source characteristics (a-d and f-h) and crystal fractionation (e and i) discrimination diagrams for the Zhoujiapuzi granite. Plots of (a) Nd/Sm vs. Ti/Eu; (b) SiO2 vs. MgO; (c) SiO2 vs. Mg#; (d) SiO2 vs. TiO2; (e) La vs. La/Yb; (f) molar Al2O3/(MgO+FeOT) vs. molar CaO/(MgO+FeOT); (g) (Na2O+K2O)/(FeOT+MgO+TiO2) vs. Na2O+K2O+FeOT+MgO+TiO2; (h) K2O/Na2O vs. Al2O3 diagrams(a after Yu et al., 2012; b-d after Wang et al., 2006; e after Gao et al., 2007; f after Altherr et al., 2000; g after Patiño Douce, 1999; h after Kamei et al., 2009)"*

8) Response to comment: Figure 1: Please check the word spelling in a), e.g., 'SOUTH

CHINA CRATON" and "CENTAL ASIAN OROGENIC BELT".

We have rechecked the spelling in fig. 1 of this MS. Sorth China Craton is corrected to South China Craton, and Central Asia Orogenic Belt is corrected to Asian rather than Asia (but not "Cental").

[Figure]

9) Response to comment: Figure 12: Please check the orientation of arrows in different diagrams and give what process the arrow refer to. Figure 14: Please give the meaning of the arrows in different diagrams.

Thank you for pointing this out. We have marked the meaning of the arrow in figures 12 and 14.

[Figure]

○ Zhoujiapuzi granite (this study)
+ Other middle-late Jurassic granitoids in Liaodong Peninsula (Wu et al., 2005a; Yang et al., 2015b, 2018; Xue et al., 2020)

Fig. 12

[Figure]

Fig. 14

○ Zhoujiapuzi granite (this study)

+ Other middle-late Jurassic granitoids in Liaodong Peninsula (Wu et al., 2005a; Yang et al., 2015b, 2018; Xue et al., 2020)

◇ Muniuhe granitic pluton (Yang et al., 2016; Wang et al., 2020)

We tried our best to improve the manuscript. We appreciate for Reviewer's warm work earnestly, and hope that the correction will meet with approval.

Once again, thank you very much for your comments and suggestions.

Sincerely,

Renyu Zeng

---

## Author Comment (AC3)

**6.4 Tectonic implications**

A large number of Early Jurassic arc-like igneous rocks occur in the northeast part of NCC- Korean Peninsula-Hida belt, which belong to the middle-high K calc-alkaline series and are characterized by enrichment in LILE and depletions in HFSE (Wu et al., 2007; Tang et al., 2018 and references therein). In addition, the Early Jurassic accretionary complexes in the eastern margin of the Eurasian continent and the Japan islands, such as the Heilongjiang complex, the Khabarovsk complex and the Mino-Tamba complex, are considered to be related to subduction (Wu et al., 2007; Tang et al., 2018 and references therein). It is generally accepted that the Paleo-Pacific slab subducted westwards in the Early Jurassic (Tang et al., 2018; Zhu and Xu, 2018).

In the middle-late Jurassic, I-type granites are dominant in the Liaodong Peninsula, such as the Zhoujiapuzi granite (this study), Heigou pluton, Gaoliduntai pluton (Wu et al., 2005a), Waling granite (Yang et al., 2015a) and Wulong granite (Yang et al., 2018). There are not A-type granites, and mantle derived magmatism is extremely rare. These granites were formed by partial melting of crustal materials without obvious contribution of mantle derived magma (Wu et al., 2005a; Yang et al., 2015b, 2018; Xue et al., 2020). In addition, WNW-ESE compression during 157-143 Ma was widespread in the Liaodong Peninsula (Yang et al., 2004; Zhang et al., 2020). It not only mylonitized the granite plutons in middle-lower crust levels, but also intensely deformed the thick sedimentary cover in the upper crust (Qiu et al., 2018; Ren et al., 2020). The high Sr/Y signature of Jurassic granites in Liaodong Peninsula do not

necessarily require crustal thickening, as mentioned before. However, based on the evidence of Cretaceous magmatism and core complexes, the Liaodong Peninsula experienced lithospheric thinning and destruction after Jurassic (Yang et al., 2007; Lin et al., 2011). Given the present crustal thickness in the region is ~30 km, the pre-Cretaceous thickness was 45-60 km – equivalent to a surface elevation of ~2–4 km, given conventional isostasy. This is also consistent with the idea that much of eastern China was a high orogenic plateau during the Mesozoic (Meng, 2003; Chu et al., 2020). Hence, Late Jurassic magmatism in the Liaodong peninsula is most likely to be related to subduction of the Paleo-Pacific plate in a mature continental arc, with crust previously thickened by compressional tectonics, related to both the oceanic subduction and the earlier Mesozoic collisions at the north and south margins of the NCC. This setting would produce the conditions required for extensive crustal melting of pre-existing basement. There is a potential resemblance to the modern arc of the Central Andes (Allmendinger et al., 1997), where crustal thickening and plateau growth has developed over the Cenozoic (Scott et al., 2018), and melting of older basement has taken place during subduction of the Nazca plate (Miller and Harris, 1989).

---

## Author Response (AR2)

Dear Johan Lissenberg and Polina Shvedko:

Thank you for your comments concerning our manuscript entitled "Whole-rock and zircon evidence for evolution of the Late Jurassic high Sr/Y Zhoujiapuzi granite, Liaodong Peninsula, North China Craton" (ID SE-2021-129). We have studied comments and suggestions carefully and have made correction. We hope meet with approval. Below the comments are response point by point and the revisions are indicated.

1. Response to comment: In the revision, you have calculated mean ages for both samples, including both core and rim data, given that these cannot be distinguished beyond analytical uncertainty (L. 261-262). Nonetheless, you write that 'the mean value for 206Pb/238U age is higher in the light-CL core' (L. 264), and that 'on a single zircon, the 206Pb/238U age of the light-CL core is older than that 266 of the dark-CL rim' (L. 265-266).
I struggle to reconcile these different statements; if cores and rims overlap within uncertainty and you treat them as a single population with an MSWD close to one, how is this compatible with different core-rime ages? Can you please clarify your reasoning on this issue and modify the manuscript accordingly?

We are sorry that some contents are not clearly stated in the MS. We try to show the relationship between the apparent age of the dark-CL rim and light-CL core, which have regular differences, but are within the error range. In the sentence "*There is a large degree of overlap between the dark-CL rim and light-CL core in terms of 206Pb/238U age although the mean value for 206Pb/238U age is higher in the light-CL core (Fig. 7e).*", "mean value" has been replaced by "average value", which can be better distinguished from "weighted mean 206Pb/238U age" in the following text. In addition, the number of analytical spots is stated in this sentence, which can emphasize that the above description is described from the perspective of all data. After the sentence "*On a single zircon, the 206Pb/238U age of the light-CL core is older than that of the dark-CL rim (Fig. 5).*", we have added " *but the two values are within the error range of the*

*in-situ LA-ICP-MS analyses (individual spot of ±3–5% relative precision, Schmitz and Kuiper, 2013).*", which can provide a basis for putting the data of light CL core and dark CL rim together to calculate the weighted average age in the following sentence. The revised content is as follows:

*"There is a large degree of overlap between the 29 spots of dark-CL rim and 32 spots of light-CL core in terms of 206Pb/238U age although the average value for 206Pb/238U age is higher in the 32 spots of light-CL core (Fig. 7e). On a single zircon, the 206Pb/238U age of the light-CL core is older than that of the dark-CL rim (Fig. 5), but the two values are within the error range of the in-situ LA-ICP-MS analyses (individual spot of ±3–5% relative precision, Schmitz and Kuiper, 2013)."*

2. Response to comment: For the next revision, please add the "Correspondence to:" to the title page of the *.pdf manuscript file.

We have added the email of the corresponding author.

*Correspondence: Renyu Zeng ([zengrenyu@126.com](zengrenyu@126.com))*

3. Response to comment: Regarding your figure #2: for the next revision, please check if your figures containing photos require a copyright statement/image credit and add it to the figures (or captions) (https://publications.copernicus.org/for_authors/manuscript_preparation.html#figurestables -> Reproduction and reuse of figures and tables). If these figures were entirely created by the authors, there is no need to add a copyright statement or credit. In that case it is important that you confirm this explicitly by email.

All the figures in Figure 2 were entirely created by the authors.

We tried our best to improve the manuscript. We appreciate for Editors' warm work earnestly, and hope that the correction will meet with approval.

Once again, thank you very much for your comments and suggestions.

Sincerely,

Renyu Zeng

---

## Author Response (AR3)

Dear editor:

We have added the following sections: *Data availability, Supplement, Author contributions, Competing interests, Acknowledgements and Financial support.*

Once again, thank you very much for your comments and suggestions.

Sincerely,

Renyu Zeng